# Two unequally redundant "helper" immune receptor families mediate *Arabidopsis thaliana* intracellular "sensor" immune receptor functions

Svenja C. Saile[1‡], Pierre Jacob[2,3‡], Baptiste Castel[4¤], Lance M. Jubic[2,3,5], Isai Salas-Gonzáles[2,3,6], Marcel Bäcker[1], Jonathan D. G. Jones[4], Jeffery L. Dangl[2,3,5,6,7,8], Farid El Kasmi[1]*

1 Center for Plant Molecular Biology, Eberhard Karls University of Tübingen, Tübingen, Germany, 2 Department of Biology, University of North Carolina at Chapel Hill, Chapel Hill, North Carolina, United States of America, 3 Howard Hughes Medical Institute, University of North Carolina at Chapel Hill, Chapel Hill, North Carolina, United States of America, 4 The Sainsbury Laboratory, Norwich Research Park, Norwich, United Kingdom, 5 Curriculum in Genetics and Molecular Biology, University of North Carolina at Chapel Hill, Chapel Hill, North Carolina, United States of America, 6 Curriculum in Bioinformatics and Computational Biology, University of North Carolina at Chapel Hill, Chapel Hill, North Carolina, United States of America, 7 Carolina Center for Genome Sciences, University of North Carolina at Chapel Hill, Chapel Hill, North Carolina, United States of America, 8 Department of Microbiology and Immunology, University of North Carolina at Chapel Hill, Chapel Hill, North Carolina, United States of America

¤ Current address: Department of Biological Sciences, National University of Singapore, Singapore
‡ These authors contributed equally to this work. JDGJ, JLD, and FEK also contributed equally to this work.
* farid.el-kasmi@zmbp.uni-tuebingen.de

**Data Availability Statement:** All relevant data are within the paper and its Supporting Information files. All necessary data and scripts to reproduce

## Abstract

Plant nucleotide-binding (NB) leucine-rich repeat (LRR) receptor (NLR) proteins function as intracellular immune receptors that perceive the presence of pathogen-derived virulence proteins (effectors) to induce immune responses. The 2 major types of plant NLRs that "sense" pathogen effectors differ in their N-terminal domains: these are Toll/interleukin-1 receptor resistance (TIR) domain-containing NLRs (TNLs) and coiled-coil (CC) domain-containing NLRs (CNLs). In many angiosperms, the RESISTANCE TO POWDERY MILDEW 8 (RPW8)-CC domain containing NLR (RNL) subclass of CNLs is encoded by 2 gene families, *ACTIVATED DISEASE RESISTANCE 1* (*ADR1*) and *N REQUIREMENT GENE 1* (*NRG1*), that act as "helper" NLRs during multiple sensor NLR-mediated immune responses. Despite their important role in sensor NLR-mediated immunity, knowledge of the specific, redundant, and synergistic functions of helper RNLs is limited. We demonstrate that the ADR1 and NRG1 families act in an unequally redundant manner in basal resistance, effector-triggered immunity (ETI) and regulation of defense gene expression. We define RNL redundancy in ETI conferred by some TNLs and in basal resistance against virulent pathogens. We demonstrate that, in *Arabidopsis thaliana*, the 2 RNL families contribute specific functions in ETI initiated by specific CNLs and TNLs. Time-resolved whole genome expression profiling revealed that RNLs and "classical" CNLs trigger similar transcriptome changes, suggesting that RNLs act like other CNLs to mediate ETI downstream of sensor NLR activation. Together, our genetic data confirm that RNLs contribute to basal resistance, are fully

every part of the RNA-Sequencing analysis are deposited in https://github.com/isaisg/helperless.

**Funding:** JDGJ and BC were supported by a core grant to the Sainsbury lab from the Gatsby foundation, and from a European Research Council grant "ImmunityByPairDesign". PJ, ISG and JLD were supported by the National Science Foundation (Grant IOS-1758400 to J.L.D.) and HHMI. J.L.D. is a Howard Hughes Medical Institute (HHMI) Investigator. L.M.J. was supported in part by a grant from the National Institute of General Medical Sciences under award 5T32 GM007092. SCS, MB and FEK were supported by core funding from the University of Tübingen, through the Deutsche Forschungsgemeinschaft [SFB/CRC1101 - project D09) and the Reinhard Frank Stiftung (Project 'helperless plant'). The funders had no role in study design, data collection and analysis, decision to publish, or preparation of the manuscript.

**Competing interests:** The authors have declared that no competing interests exist.

**Abbreviations:** Ac2V, *Albugo candida* race 2; ADR1, *ACTIVATED DISEASE RESISTANCE 1*; BTH, benzothiadiazole; CC, coiled-coil; CDS, coding sequence; cfu, colony-forming units; CNL, CC domain-containing NLR; Col-0, Columbia-0; DI, disease index; dpi, days post infection; DEG, differentially expressed gene; EDS1, *Enhanced Disease Susceptibility 1*; ETI, effector-triggered immunity; FDR, false discovery rate; GO, gene ontology; Hpa, *Hyaloperonospora arabidopsidis*; hpi, hours post infection; HR, hypersensitive response; JA, jasmonic acid; LRR, leucine-rich repeat; lsd1, *lesion simulating disease 1*; NB, nucleotide binding; NLR, NB LRR receptor; NRG1.1, *N REQUIREMENT GENE 1.1*; $OD_{600}$, optical density at 600 nm; PAD4, *Phytoalexin Deficient 4*; PAMP, pathogen-associated molecular pattern; Pf0-1, *Pseudomonas fluorescens 0–1*; PM, plasma membrane; PRR, PAMP recognition receptor; Psm, *Pseudomonas syringae* pv. *maculicola*; Pst, *Pseudomonas syringae* pv. *tomato*; PTI, pattern-triggered immunity; RK, receptor kinase; RLP, receptor-like protein; RNL, RPW8-CC domain containing NLR; Roq1, RECOGNITION OF XOPQ 1; ROS, reactive oxygen species; RPM1, RESISTANCE TO P. SYRINGAE PV MACULICOLA 1; RPP1, RECOGNITION OF PERONOSPORA PARASITICA 1; RPS2, RESISTANT TO P. SYRINGAE 2; RPS4, RESISTANT TO P. SYRINGAE 4; RPS5, RESISTANT TO P. SYRINGAE 5; RPS6, RESISTANT TO P. SYRINGAE 6; RPW8, RESISTANCE TO POWDERY MILDEW 8; RRS1, RESISTANT TO RALSTONIA SOLANACEARUM 1; SA, salicylic acid; SAG101, *Senescence Associated*

required for TNL signaling, and can also support defense activation during CNL-mediated ETI.

# Introduction

Plant defense responses, once initiated, thwart attacking and invading pathogens via multiple mechanisms [1,2]. Microbial pathogens can trigger a first defense response upon detection by plasma membrane (PM)-localized leucine-rich repeat (LRR) receptor kinases (RKs) or receptor-like proteins (RLPs). These receptors perceive conserved pathogen-associated molecular patterns (PAMPs) or danger signals from diverse pathogens and initiate a broad range of immune responses, collectively called pattern-triggered immunity (PTI) [3,4]. PTI effectively inhibits non-host-adapted pathogens and also contributes to resistance to host-adapted pathogens. Plant pathogens deliver virulence effectors into plant cells by a variety of mechanisms to dampen PTI [5]. In turn, plants evolved the ability to recognize effectors or their action on host targets, thereby initiating a second tier of the immune system, effector-triggered immunity (ETI). ETI involves strong activation of defense mechanisms and is often associated with a type of cell death at the site of infection termed the hypersensitive response (HR) [6,7]. In nearly all cases, ETI is mediated by intracellular immune receptors called nucleotide-binding (NB) LRR receptors (NLRs). NLRs are modular proteins, typically exhibiting a C-terminal LRR domain, a central nucleotide-binding (NB) domain, and any of a small variety of N-terminal domains [8,9]. In plants, 2 major types of NLRs have been described which differ in their N-terminal domains. The Toll/interleukin-1 receptor resistance (TIR) domain-containing NLRs (TNLs) and the coiled-coil (CC) domain-containing NLRs (CNLs) can directly or indirectly sense the presence of pathogen effectors [9]. Thus, they are usually designated "sensor" NLRs. Further, there is a unique subclade of CNLs that exhibit an atypical CC-R N-terminal domain sequence-related to the resistance protein RESISTANCE TO POWDERY MILDEW 8 (RPW8), therefore also termed RNLs. RNLs are required for the function of many sensor NLRs and are thus also referred to as helper NLRs [10–15]. CNL and TNL activation by effectors may involve NLR oligomerization—the formation of a so-called resistosome—that is required for NLR function in immunity, as was shown for the *Arabidopsis thaliana* CNL HOP-Z-ACTIVATED RESISTANCE 1 (ZAR1) [16–19].

RNLs form an evolutionarily conserved clade of NLRs present in most land plants that share a unique NB domain in addition to the RPW8-like CC-R domain [20,21]. *A. thaliana* has 2 subfamilies of RNLs, ADR1s and NRG1s. The ADR1 family includes ACTIVATED DISEASE RESISTANCE 1 (ADR1) and paralogs ADR1-LIKE 1 (ADR1-L1), ADR1-L2, and ADR1-L3, while the NRG1 family includes N REQUIREMENT GENE 1.1 (NRG1.1) and paralogs NRG1.2 and NRG1.3 [14,22,23]. We will refer to the *NRG1* genes according to The Arabidopsis Information Resource (TAIR; https://www.arabidopsis.org/index.jsp) nomenclature as *NRG1.1*, *NRG1.2*, and *NRG1.3* hereafter. These correspond to *NRG1A*, *NRG1B*, and *NRG1C* as used, e.g., in work by Wu and colleagues [24]. *ADR1-L3* and *NRG1.3* could encode N-terminally truncated proteins but have no documented function in immunity, as recently demonstrated for *NRG1.3* [11].

The study of RNLs unveils complex genetic interactions. *ADR1-L1* and *ADR1-L2* were first studied for their crucial role in the run-away-cell death phenotype initiated by the application of the salicylic acid (SA) analogue benzothiadiazole (BTH) to the *lesion simulating disease 1* (*lsd1*) mutant background [14]. Both the *adr1-L1* and the *adr1-L2* mutation suppressed the *lsd1* runaway-cell-death. Further, the double heterozygous *ADR1-L1/adr1-L1 ADR1-L2/*

*Gene* 101; SAR, systemic acquired resistance; T3SS, type III secretion system; T-DNA, transfer DNA; TAIR, The Arabidopsis Information Resource; TIR, Toll/interleukin-1 receptor resistance; TNL, TIR domain-containing NLR; WRR4, WHITE RUST RESISTANCE 4; ZAR1, HOPZ-ACTIVATED RESISTANCE1.

*adr1-L2* mutant also suppressed the *lsd1* runaway-cell-death phenotype. Thus *ADR1-L1* and *ADR1-L2* constitute a rare case of non-allelic non-complementation. This suggested that both genes contribute quantitatively to the cell death phenotype [25]. *ADR1* was not involved in this phenotype but was found to be redundantly involved, with *ADR1-L1* and *ADR1-L2*, in SA accumulation and ETI activation downstream of some sensor NLRs—including examples of both CNLs and TNLs [14,26]—and was also found to elevate disease resistance when overexpressed [23].

NRG1 was first discovered in *Nicotiana benthamiana* via its requirement for ETI induced by the sensor TNL protein N [22]. In *A. thaliana* Columbia-0 (Col-0) plants, 2 tightly linked functional copies of *NRG1* exist, *NRG1.1* and *NRG1.2*. The application of CRISPR-Cas9 technology allowed the generation of *A. thaliana nrg1.1 nrg1.2* double [13,27] and *nrg1.1 nrg1.2 nrg1.3* triple mutants [11]. Recently, a "*helperless*" mutant lacking all RNLs was constructed by combining the *nrg1* double or triple CRISPR-Cas9 alleles with a preexisting *adr1 adr1-L1 adr1-L2* triple mutant (hereafter *adr1 triple*), [11,14,27]. Using the combinatorial *nrg1* mutants, as well as a *N. benthamiana nrg1* mutant, it was determined that NRG1s are broadly required for TNL function (N, Roq1, RPS4/RRS1, RPP1, among others [10–13]).

The immune phenotype of the *helperless* plant was only examined for its effect on 2 sensor TNLs, RPS4/RRS1 and RPS6, as well as in basal defense against *Pseudomonas syringae* pv. *maculicola* (*Psm*) ES4326 [11,13,27]. While *nrg1.1 nrg1.2* did not show significant disease susceptibility, it did enhance the susceptibility phenotype of the *adr1 triple* mutant. Thus, it is likely that an unequal genetic redundancy between *ADR1s* and *NRG1s* has masked the true function of NRG1s and the importance of their helper function during ETI.

We systematically compared the *adr1 triple* mutant, the *nrg1.1 nrg1.2* double mutant, and a newly generated *helperless* mutant to characterize RNL function and the genetic interactions between the *ADR1* and the *NRG1* families. We demonstrate that the 2 RNL subfamilies function redundantly (with some specificity) in disease resistance to biotrophic and hemibiotrophic virulent and avirulent pathogens, ETI-induced gene expression, and HR. We describe an important role for the ADR1s in ETI and basal resistance. Further, we note a partial subfunctionalization or specialization of the 2 RNL families, specifically during sensor NLR-triggered ETI and in resistance against the necrotrophic fungal pathogen *Alternaria brassicicola*. Infection assays with a coronatine-deficient *Pseudomonas syringae* pv. *tomato* (*Pst*) DC3000 cor- show that both RNL families are involved to different extents in regulating SA-related pathways targeted by the jasmonic acid (JA) analogue coronatine.

Finally, time-resolved whole-genome expression profiling revealed that RNL-regulated genes are also CNL regulated and vice versa. Overall, we propose that ADR1s and NRG1s are required downstream of all sensor TNLs to activate ETI and that both RNL families are required to support the activation of—in the case of some CNLs—an orthodox CNL-initiated ETI.

## Results

### Redundant functions of ADR1 and NRG1 subfamilies in TNL-mediated disease resistance

Although it has been reported that the ADR1s and NRG1s function downstream of some TNLs [11–14,27], we lack a detailed comparison of the different requirements for the ADR1s and NRG1s during TNL-mediated ETI. To describe the specific and redundant immune functions of *ADR1* and *NRG1* subfamilies, we generated a *helperless* mutant by CRISPR-Cas9-mediated knock-out of the 2 full-length *NRG1* genes (*NRG1.1* and *NRG1.2*) in the *adr1 triple* transfer-DNA (T-DNA) line [14]. The *adr1 triple*, *nrg1.1 nrg1.2* double [13] and the

"*helperless*" pentuple mutants were infected with the bacterial pathogen *Pst* DC3000 expressing the effector AvrRps4. AvrRps4-induced ETI requires the cooperative function of the TNLs RPS4 and RRS1 [28]. Three days post infection (dpi), the *nrg1.1 nrg1.2* double mutant displayed wild-type–like resistance (Fig 1A). In contrast however, RPS4/RRS1-mediated resistance was significantly compromised in the *adr1 triple* and *helperless* mutant (Fig 1A). Interestingly, the *helperless* mutant was much more susceptible to *Pst* DC3000 *AvrRps4* infection than either the *adr1 triple* or the *rrs1a rrs1b* control plants (Fig 1A). Our results are consistent with published observations (S1 Table; [11,27]) and indicate that *ADR1s* and *NRG1s* function in an unequally redundant manner in RPS4/RRS1-triggered resistance. This type of genetic interaction occurs when 2 related genes or gene families are engaged in the process of sub-functionalization after gene duplication [29]. In this situation, one of the copies has lost most of the ancestral function, and its loss of function can be fully compensated by the other gene/gene subfamily. However, it can still contribute to the ancestral function, and double mutants have an enhanced phenotype compared to either single mutant. This observation raised the concern that the definition of functions of *NRG1s* in defense may have been effectively hidden by the *ADR1s*. The "residual" resistance of *rrs1a rrs1b* against *Pst* DC3000 *AvrRps4* compared to the *helperless* mutant suggests that additional weak recognition events might be impaired in the *helperless* mutant. This is similar to the phenotype of the *enhanced disease susceptibility 1* (*eds1*) mutant in response to *Pst* DC3000 *AvrRps4* infections. Here, *eds1-12* is also more susceptible than an *rrs1a rrs1b* double mutant [13,30], thus indicating that the *helperless* mutant phenocopies an *eds1* mutant during ETI mediated by RPS4/RRS1.

To further test whether the unequal redundancy between *ADR1s* and *NRG1s* extends beyond RPS4/RRS1-mediated resistance, *adr1 triple*, *nrg1.1 nrg1.2* double, and *helperless* mutants were challenged with the oomycete *Hyaloperonospora arabidopsidis* (*Hpa*) isolate Cala2. Resistance against *Hpa* Cala2 is mediated by the sensor TNL RPP2 in Col-0 [31]. While *nrg1.1 nrg1.2* was as resistant as wild type at 6 dpi, both the *adr1 triple* and *helperless* mutants exhibited increased susceptibility, with the *helperless* mutant being much more susceptible than *adr1 triple* (Fig 1B). The RPP2-dependent immune response in the *helperless* mutant to *Hpa* Cala2 infection was as severely affected as in *eds1-12* and the sensor NLR mutant *rpp2a* (Fig 1B). Together, these data provide another example of unequal redundancy of the 2 RNL families during a TNL-mediated immune response.

Infection of the 3 combinatorial RNL mutants with another obligate biotrophic oomycete pathogen, *Albugo candida* race 2V (Ac2V), which is recognized by the TNL WHITE RUST RESISTANCE 4A (WRR4A) and an unknown recessive disease resistance gene in Col-0 [13,32,33], demonstrated that *NRG1* and *ADR1* families can be fully redundant (Fig 1C). Col-0 and both the *adr1 triple* and *nrg1.1 nrg1.2* mutants were resistant to Ac2V, whereas the *helperless* plant was as sensitive as *eds1-12*. This indicates that *ADR1s* and *NRG1s* are redundantly required for WRR4A-mediated resistance.

Our combined results suggest that the 2 RNL families can act in a fully or unequally redundant manner in TNL-triggered immunity, likely depending on the sensor TNL activated during the infection.

## Specific functions of the RNL subfamilies in CNL- and TNL-triggered ETI

Recent studies suggested that *ADR1s* and *NRG1s* have specific, nonredundant functions in sensor NLR-mediated immunity [11,13,27]. The *NRG1* family was suggested to function specifically in HR/cell-death induction after the activation of RPS4/RRS1 or during transient overexpression of auto-active full-length TNLs and TIR domains in *N. benthamiana* [13,27]. Additionally, *A. thaliana* NRG1.1 and NRG1.2 are required for some but not all TNL-mediated

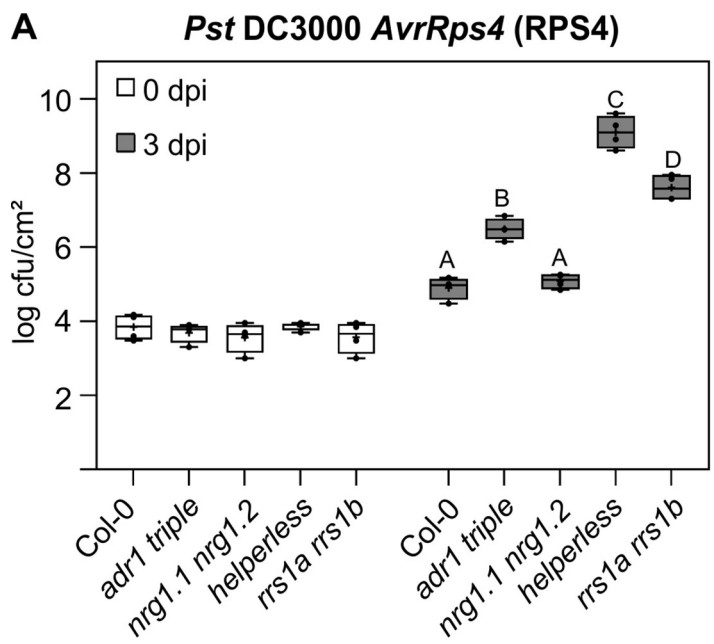

**A** *Pst* DC3000 *AvrRps4* (RPS4)

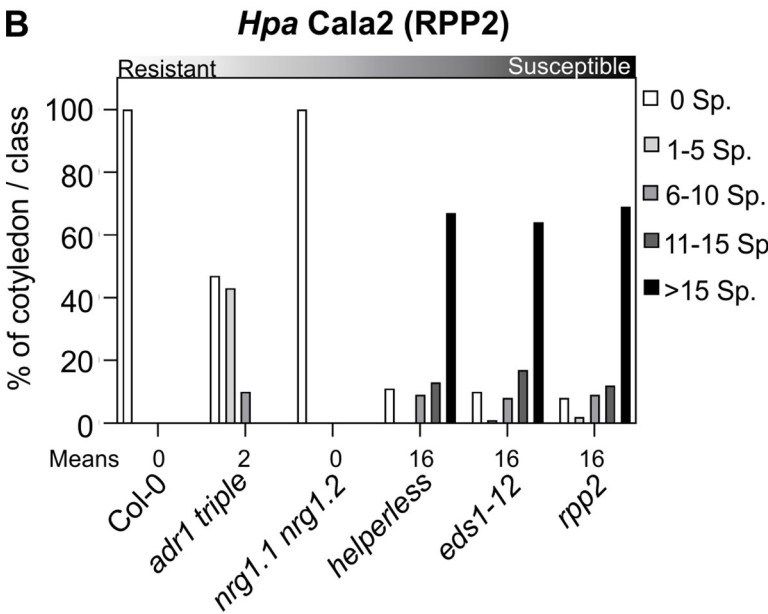

**B** *Hpa* Cala2 (RPP2)

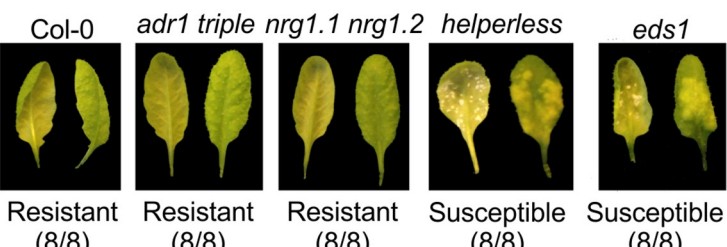

**C** *Albugo candida* race 2V (WRR4A)

**Fig 1. Redundant functions of ADR1 and NRG1 subfamilies in TNL-mediated resistance.** (A) Six-week-old plants were hand-infiltrated with *Pst* DC3000 *AvrRps4* ($OD_{600}$ = 0.001), and bacterial growth was assessed at 0 and 3 dpi. Box limit represents upper and lower quartile; maximum and minimum values are displayed in whiskers. The middle line shows the median, the cross the mean $cfu/cm^2$. Dots represent 4 technical replicates (leaf discs) in one experiment (biological replicate). Experiment was done 3 times with similar results. Letters indicate statistically significant differences following ANOVA with Tukey's test ($\alpha$ = 0.05). (B) Ten-day-old seedlings were inoculated with *Hpa* Cala2. Sporangiophores per cotyledon were counted at 5 dpi. Cotyledons were classified as supporting no sporulation (0 Sp./cotyledon), light sporulation (1–5 and 6–10), medium sporulation (11–15), or heavy sporulation (>15). Two independent experiments were performed with an average of 100 cotyledons counted per genotype. Means of Sp./cotyledon for each genotype are noted below. Standard deviations of means for each genotype are as follows: Col-0 ± 0, *adr1 triple* ± 0.25, *nrg1.1 nrg1.2* ± 0, *helperless* ± 0.68, *eds1-12* ± 0.68 and *rpp2a* ± 0.65. Calculations for statistically significant differences following ANOVA with Tukey's test ($\alpha$ = 0.05) are provided in S1 Data. (C) Three- to five-week-old plants were spray inoculated with Ac2V. Plants were phenotyped at 12 dpi. Abaxial and adaxial photographs of the same leaf are shown. Numbers indicate the number of individual plants showing a similar phenotype from the number of plants tested. NLRs activated in infection experiments shown in A–C are indicated in parenthesis. Underlying numerical data are provided in S1 Data. Ac2V, *Albugo candida* race 2V; cfu, colony-forming units; Col-0, Columbia-0; dpi, days post infection; *Hpa*, *Hyaloperonospora arabidopsidis*; NLR, nucleotide-binding leucine-rich repeat receptor; $OD_{600}$, optical density at 600 nm; Pst, *Pseudomonas syringae* pv. *tomato*; Sp., Sporangiophores; TNL, Toll/interleukin-1 receptor resistance domain-containing NLR.

autoimmune phenotypes [11–13,34]. The *ADR1s* are required for full ETI mediated by effector-triggered RPS2 and RPP4 [11,13]. We investigated the possibility that unequal genetic redundancy may have hidden some functions of *NRG1s* and *ADR1s* by including the *helperless* plant in all of our *Pst*, *Pseudomonas fluorescens 0–1* (*Pf0-1*; "Effector To Host Analyzer" strain derived from *P. fluorescens* [35]), and *Hpa* infection assays (Fig 2). We confirmed the specific requirement of the *ADR1* family in both RPS2- and RPP4-mediated resistance, as there was no significant increase in susceptibility in the *helperless* mutant compared to the *adr1 triple* in our *Pst* DC3000 *AvrRpt2* and *Hpa* Emwa1 infection assays (Fig 2A and 2B). We observed a slight, but consistent, increase of susceptibility in the *rps2* mutant compared to the *adr1 triple* and *helperless* mutants, suggesting a residual RNL-independent function of RPS2. We did not observe a significant defect in RPS2- and RPP4-mediated resistance in *nrg1.1 nrg1.2* (Fig 2A and 2B). Similarly, we found that resistance to *Pst* DC3000 *AvrPphB* (which activates the CNL RESISTANT TO P. SYRINGAE 5 [RPS5] [36]) was not affected in *nrg1.1 nrg1.2*, while both *adr1 triple* and *helperless* mutants showed an enhanced susceptibility (Fig 2C). The sensitivity phenotype was similar in *adr1 triple* and *helperless* mutants, suggesting that the ADR1 family is specifically required to mediate the defense response to *Pst* DC3000 *AvrPphB*. We next examined the effect of the different RNL mutants on the HR induction upon effector-mediated RPS4 and RPS2 activation. RPS4 activation led to an NRG1-dependent HR (Fig 2D), whereas only the ADR1s were required for HR after RPS2 activation (Fig 2E). There was no visible HR in the *adr1 triple* or *helperless* mutants at 10 hours post infection (hpi) (Fig 2E; [14]). A delayed RPS2-mediated HR was, however, visible at 24 hpi regardless of the genotype, showing that ADR1s are required for the timely activation of RPS2-triggered HR and RPS2-mediated disease resistance (Fig 2A and 2E). Our results suggest that the ADR1s support ETI mediated by the CNLs RPS2 and RPS5, but are required for the full ETI triggered by the TNL RPP4. NRG1s are the RNLs mediating HR triggered by the activation of RPS4/RRS1 and do not have any obvious function during RPS2-mediated HR and disease resistance.

## Unequal redundancy of the RNL families in basal resistance

Next, we analyzed whether the 2 RNL families have specific or redundant functions during basal resistance or PTI. Basal resistance is defined as the resistance that is activated by PTI minus the consequences of effector-mediated suppression of PTI, but including any residual weak ETI [2]. ADR1s are required for both basal resistance against virulent pathogens and SA

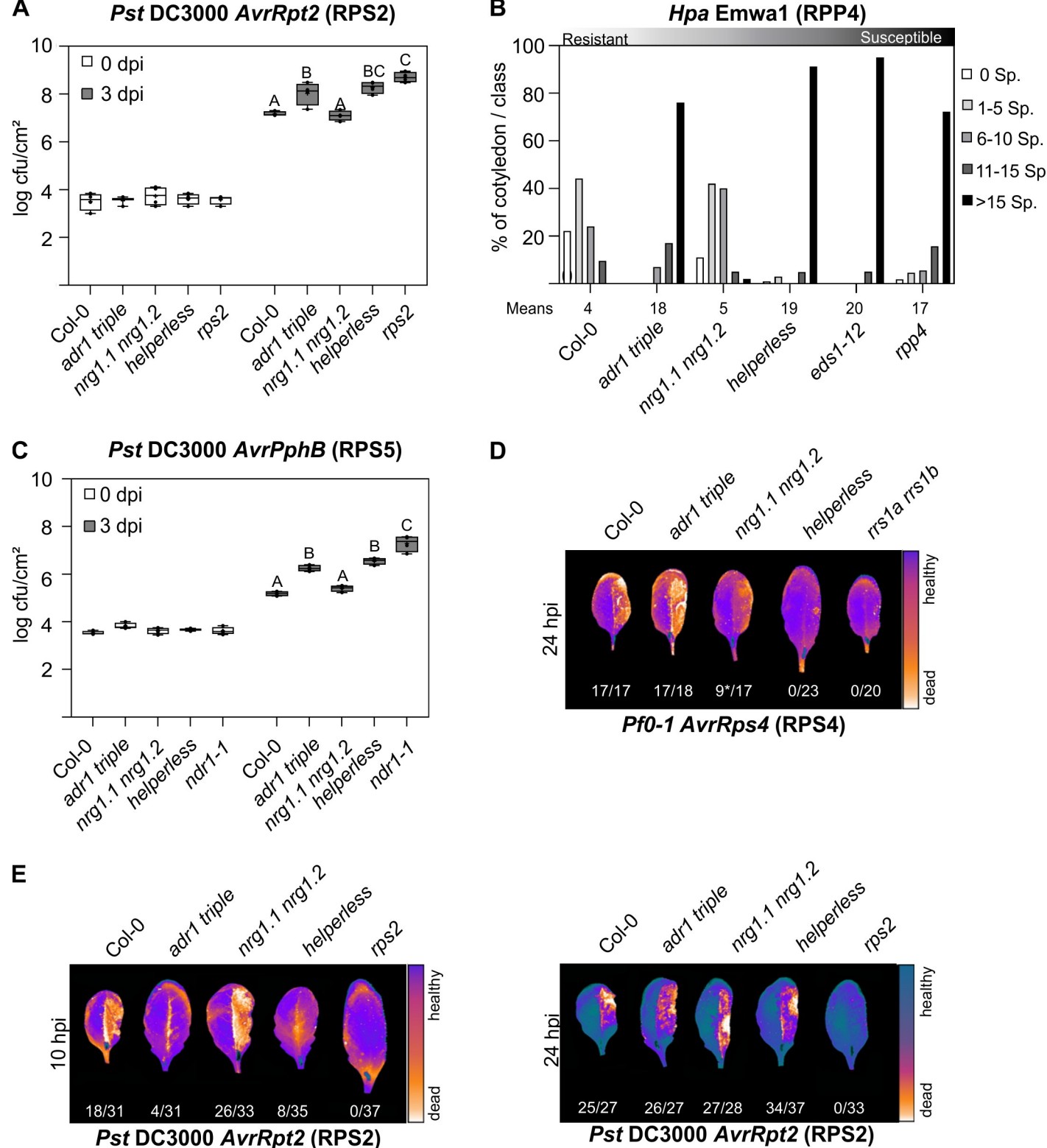

**Fig 2. Specific functions of ADR1 and NRG1 subfamilies during ETI.** (A, C) Six-week old plants were hand-infiltrated with (A) *Pst* DC3000 *AvrRpt2* ($OD_{600}$ = 0.001) or (C) *Pst* DC30000 *AvrPphB* ($OD_{600}$ = 0.001), and bacterial growth was assessed at 0 and 3 dpi. Box limit represents upper and lower quartile; maximum and minimum values are displayed in whiskers. The middle line shows the median, the cross the mean cfu/cm². Dots represent 4 technical replicates (leaf discs) in one experiment (biological replicate). Experiment was done 3 times with similar results. Letters indicate statistically significant differences following ANOVA with Tukey's

test ($\alpha$ = 0.05). (B) Ten-day-old seedlings were inoculated with *Hpa* Emwa1. Sporangiophores per cotyledon were counted at 5 dpi. Cotyledons were classified as supporting no sporulation (0 Sp./cotyledon), light sporulation (1–5 and 6–10), medium sporulation (11–15), or heavy sporulation (>15). Two independent experiments were performed with an average of 100 cotyledons counted per genotype. Two independent experiments were performed with an average of 100 cotyledons counted per genotype. Means of sporangiophores/cotyledon for each genotype are noted below. Standard deviations of means for each genotype are as follows: Col-0 ± 0.77, *adr1 triple* ± 0.36, *nrg1.1 nrg1.2* ± 0.39, *helperless* ± 0.36, *eds1-12* ± 0.13, and *rpp4* ± 0.5. Calculations for statistically significant differences following ANOVA with Tukey's test ($\alpha$ = 0.05) are provided in S1 Data. (D, E) The right leaf half of 6-week-old plants was hand-infiltrated with (D) *Pf0-1 AvrRps4* ($OD_{600}$ = 0.2) or (E) *Pst* DC3000 *AvrRpt2* ($OD_{600}$ = 0.1). The Typhoon laser scanner was used to detect autofluorescence of dead leaf tissue at indicated time points. Representative leaves shown in a false color scale (black to blue: healthy leaf tissue, orange to white: dead leaf tissue). Numbers indicate the amount of leaves showing HR out of the total number of leaves analyzed. Asterisk in D indicates weak HR. NLRs activated in infection experiments shown in A–E are indicated in parentheses. Underlying numerical data are provided in S1 Data. cfu, colony-forming units; Col-0, Columbia-0; dpi, days post infection; ETI, effector-triggered immunity; *Hpa*, *Hyaloperonospora arabidopsidis*; hpi, hours post infection; HR, hypersensitive response; NLR, nucleotide-binding leucine-rich repeat receptor; $OD_{600}$, optical densitiy at 600 nm; *Pf0-1*, *Pseudomonas fluorescens 0–1*; *Pst*, *Pseudomonas syringae* pv. *tomato*; Sp., Sporangiophores.

accumulation during PTI [11,14]. Recently, Wu and colleagues showed an increased susceptibility to *Psm* ES4326 infections due to the loss of *NRG1s* in the *adr1 triple* mutant context, but not in the wild-type context [11]. This suggests that NRG1s might have a function in basal resistance. To further investigate this hypothesis, we infected *adr1 triple*, *nrg1.1 nrg1.2*, and *helperless* mutants with the virulent pathogen *Pst* DC3000 carrying an empty cloning vector (hereafter, EV). We did not observe any significant effect on *Pst* DC3000 EV growth in the *nrg1.1 nrg1.2* mutant, whereas the *adr1 triple* and the *helperless* plants were more susceptible than Col-0, comparable to *eds1-12* (Fig 3A). This result suggests a specific function for the *ADR1s* in basal resistance against *Pst* DC3000 in Col-0, in line with the proposition of collective weak ETI triggered by the recognition of some effectors [7].

Considering the involvement of *NRG1s* in basal defense against *Psm* ES4326 [11], we reasoned that the impact of *nrg1.1 nrg1.2* might be epistatic to some virulence factor of *Pst* DC3000. We took advantage of the coronatine-deficient mutant, *Pst* DC3000 cor-, to further determine the function of *NRG1s* in basal immunity [37]. Coronatine antagonizes SA signaling, which is an important feature of immune signaling by *ADR1s* [14,38,39]. We observed enhanced bacterial growth in the *adr1 triple* and further enhanced growth in the *helperless* mutant, while no difference between *nrg1.1 nrg1.2* and Col-0 was detected. This observation indicates that the *NRG1s* play a role in basal resistance against *Pst* DC3000 that is visible only in an *adr1 triple* mutant background and when the virulence function of coronatine is removed (Fig 3B).

To test whether RNLs are also involved in PTI, we infected the *adr1 triple*, *nrg1.1 nrg1.2*, and *helperless* mutants with the type III secretion system (T3SS)-deficient mutant *Pst* DC3000 ΔhrcC. This mutant is impaired in delivering type-III effectors into the host plant, and therefore its growth is severely restricted by the strong activation of PTI responses that are not suppressed by effector functions [40]. None of the infected RNL mutants showed an enhanced bacterial growth of *Pst* DC3000 ΔhrcC compared to Col-0 (Fig 3C), suggesting that RNLs do not play a critical role during PTI triggered by this disarmed pathogen. This is further supported by our analysis of the flagellin-22 peptide (flg-22) induced reactive oxygen species (ROS) burst, which is one of the best-characterized PTI responses [41]. We did not observe any statistically significant differences in any RNL mutant compared to Col-0 (S1 Fig), suggesting that RNL function is not essential for this PTI response.

In summary, Fig 3A to 3C demonstrates that *ADR1s* and *NRG1s* do not play a critical role during PTI, but are unequally redundant for basal resistance against virulent *Pseudomonas*. The contribution of the *NRG1s* is only visible in the absence of the prevalent *ADR1s* and in defense mediated against *Pst* DC3000 cor-. Retention of *Pst* DC3000 growth restriction in *nrg1.1 nrg1.2* mutants may be due to the antagonistic effect of coronatine on SA signaling.

SA-dependent defense responses are well known to be required for resistance against biotrophic pathogens, whereas necrotrophic pathogens are better resisted by JA-dependent

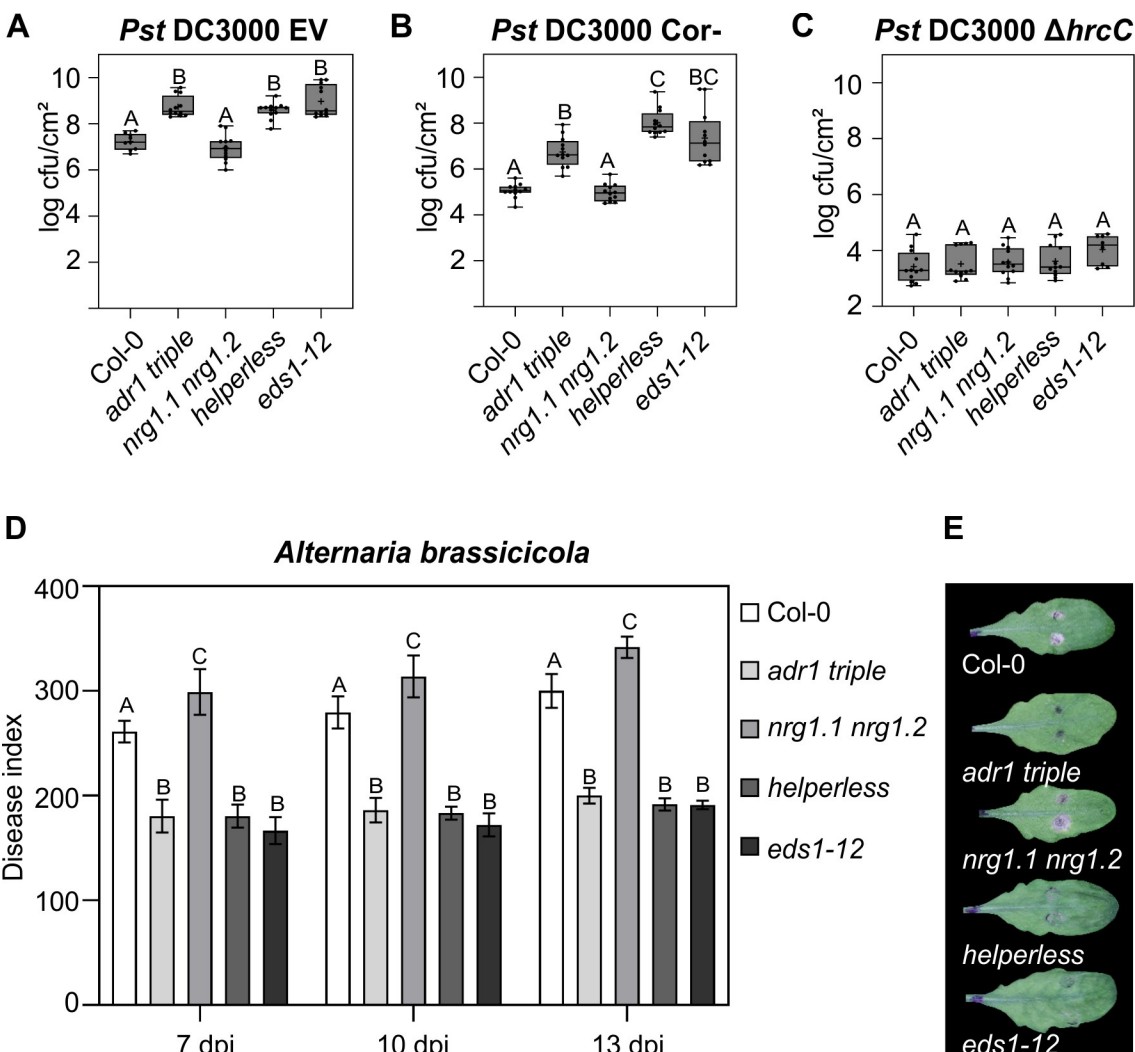

**Fig 3. Unequally redundant and specific functions of RNLs during basal resistance and resistance against a necrotrophic pathogen.** (A, B, C) Six-week-old plants were hand-infiltrated with (A) *Pst* DC3000 EV ($OD_{600}$ = 0.001), (B) *Pst* DC3000 cor- ($OD_{600}$ = 0.002), or (C) Pst DC3000 $\Delta hrcC$ ($OD_{600}$ = 0.002), and bacterial growth was assessed at 3 dpi. Dots represent 12 data points (3 biological replicates and 4 technical replicates). Box limit represents upper and lower quartile; maximum and minimum values are displayed in whiskers. The middle line shows the median, the cross the mean cfu/cm². Letters indicate statistically significant differences following ANOVA with Tukey's test ($\alpha$ = 0.05). (D) 5.5-week-old plants were inoculated with $1 \times 10^6$ spores/mL *A. brassicicola*, and disease symptoms were monitored at 7, 10, and 13 dpi. DIs are shown as mean ± SEM of at least 35 replicates of 2 independent experiments. Letters indicate statistically significant differences at one time point following ANOVA with Tukey's test ($\alpha$ = 0.05). (E) Pictures of representative leaves inoculated with two 5 µL droplets of *A. brassicicola* spores were taken 13 dpi. Underlying numerical data are provided in S1 Data. cfu, colony-forming units; DI, disease index; dpi, days post infection; EV, empty vector; *hrcC*, *HR and pathogenicity gene C*; $OD_{600}$, optical density at 600 nm; *Pst*, *Pseudomonas syringae* pv. *tomato*.

immune responses [42]. The prevalent function of the ADR1s in the regulation of the SA-signaling pathway [13,14] and the antagonistic relationship of the SA and JA pathways [43] prompted us to analyze the resistance of the RNL mutants to the necrotrophic pathogen *Alternaria brassicicola*. Both the *adr1 triple* and *helperless* mutants restricted fungal growth better than wild-type Col-0, phenocopying *eds1-12* (Fig 3D and 3E). The *nrg1.1 nrg1.2* plants were slightly, but consistently, more susceptible to *A. brassicicola* than Col-0. The enhanced resistance in *adr1 triple* and *eds1-12* is most likely due to a loss of the antagonistic function of SA signaling on the JA-dependent immune response in these mutants.

These data confirm a major role of the ADR1s and EDS1 in the positive regulation of SA signaling and accumulation and further support the idea that the ADR1s and EDS1 (and PHYTOALEXIN DEFICIENT 4 [PAD4]) function together or in parallel in plant immunity.

## RNL-independent functions during CNL-triggered ETI

We and others showed a requirement of both RNL families in TNL-mediated ETI and that *ADR1*s function in RPS2- and RPS5-mediated ETI (Fig 2) [11–14,27,34]. However, assessment of the 2 RNL families for other CNL-mediated ETI responses is limited. RESISTANCE TO P. SYRINGAE PV MACULICOLA 1 (RPM1)-mediated ETI is independent of *ADR1*s, and a very weak effect on the timing of RPM1-mediated HR was observed in *nrg1.1 nrg1.2* [13,14]. In addition, RPS5-mediated HR was not compromised in *nrg1.1 nrg1.2* [13]. In order to determine the requirement of RNLs in CNL-mediated ETI, we analyzed bacterial growth restriction in RNL mutants upon infection with *Pst* DC3000 expressing AvrRpm1 or HopZ1a (S2 Fig). Bacterial growth restriction upon *Pst* DC3000 *AvrRpm1* or *HopZ1a* infiltration was comparable to Col-0 in all RNL mutants analyzed (S2A and S2B Fig). This demonstrates that RNL presence is not required for RPM1- or ZAR1-mediated bacterial growth restriction.

We further tested whether effector-activated RPM1, ZAR1, and RPS5 require RNLs for HR induction. We analyzed visual HR symptoms in all 3 Arabidopsis RNL mutants at 6 hpi (for RPM1), 22 hpi (for RPS5), and 24 hpi (for ZAR1) with *Pst* DC3000 expressing the respective effectors (S2C–S2E Fig). Additionally, we performed ion leakage assays on all RNL mutants for both *Pf0-AvrRpm1* and *Pst* DC3000 *AvrRpm1* infections (S2G and S2H Fig). Similar to our growth restriction data, we did not observe any effect on HR timing or strength in the RNL mutants, confirming previously published results [13,14].

Our data demonstrate that RPM1- and ZAR1-mediated ETI responses (HR induction and bacterial growth restriction) and RPS5-mediated HR do not require RNLs. It would be interesting to further determine whether this holds true for other CNLs or whether our results are specific for CNLs known (RPM1 and RPS5 [44,45]; ZAR1 [24]) to act at the PM.

## RNL requirement in transcriptional reprogramming during PTI

We further investigated the function of RNLs in transcriptional reprogramming during *Pf0-1*-induced PTI and ETI, using time-resolved transcriptomics. We subjected Col-0, *adr1 triple*, *nrg1.1 nrg1.2*, and *helperless* mutants to infections with *Pf0*-EV (RNL-[in]dependent PTI), *Pf0-AvrRps4* (fully RNL-dependent ETI + PTI), *Pf0-AvrRpt2* (partial RNL-dependent ETI + PTI), or *Pf0-AvrRpm*1 (RNL-independent ETI + PTI). Four leaf discs from 4 different plants at time 0 (before treatment), 0.5 hpi (PTI induction), 4 hpi (early ETI), and 8 hpi (late ETI) were used for RNA extraction, and mRNAs were sequenced as single-end, 50-bp reads, yielding approximately 5 million reads per sample. Two independent samples were gathered within each experiment, and the experiment was performed 3 times.

A large number of genes were found differentially expressed during at least one treatment and time point with a false discovery rate (FDR)-adjusted $p < 0.05$ and a fold change > 2 (Fig 4A). Principal component analysis showed the main factors affecting gene expression were (1) treatment time, (2) treatment type, and (3) plant genotype (Fig 4B). At 0.5 hpi, every treatment triggered changes in expression of mostly the same genes compared to time 0 (Fig 4A and 4B), consistent with the fact that type III effectors are generally not delivered at this stage [46]. The differences between genotypes were not clearly distinguishable at 0.5 hpi. The majority of gene expression changes in the RNL mutants resembled that of Col-0 during *Pf0*-EV infection at all 3 time points analyzed (Fig 4A, 4B and 4C). We conclude that the contribution of RNLs to overall gene expression regulation during PTI was very limited.

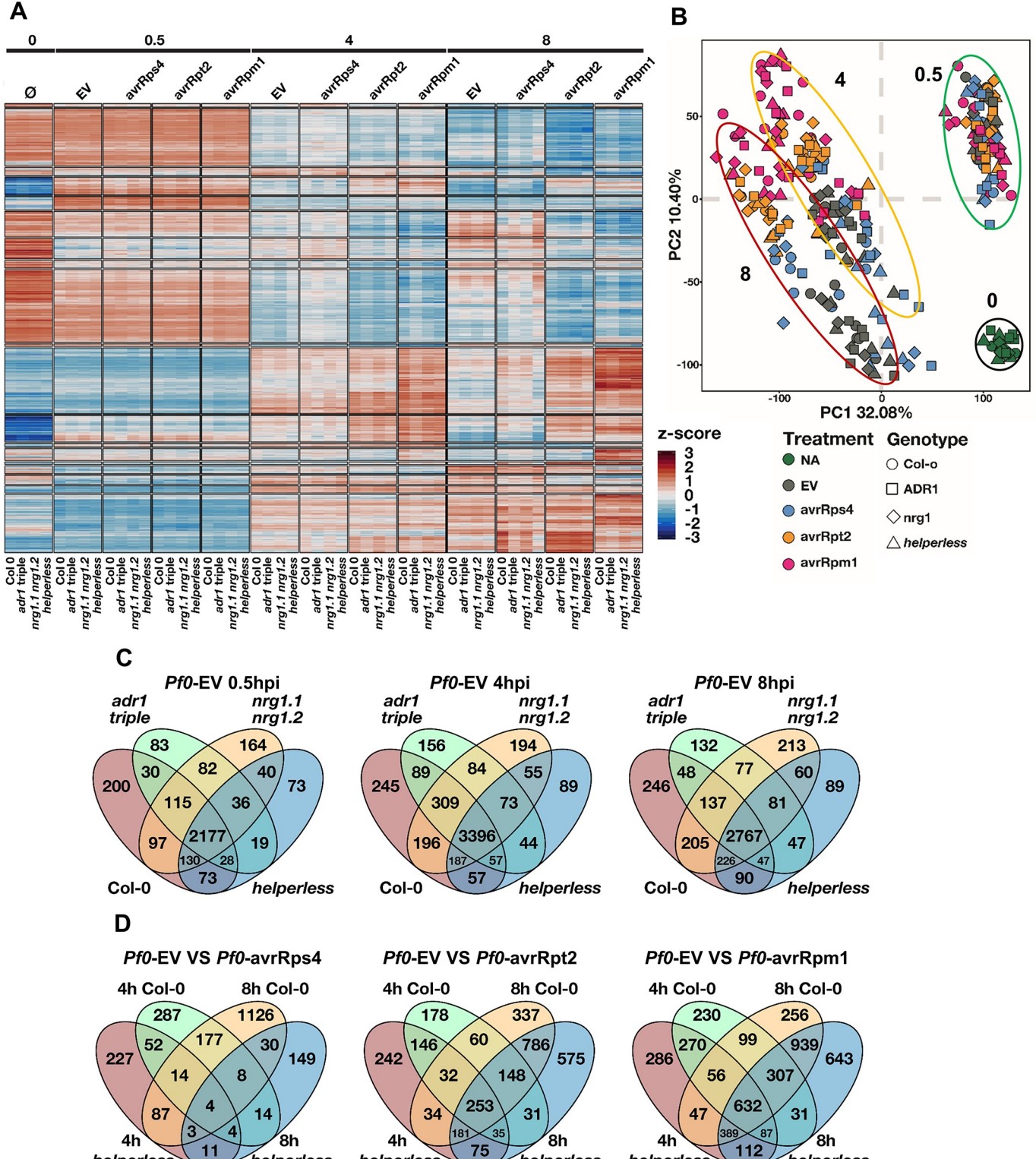

**Fig 4. Analysis of RNL-dependent and -independent transcriptional changes.** Six-week-old Col-0, *adr1 adr1-L1 adr1-L2* (*adr1 triple*), *nrg1.1 nrg1.2*, or *adr1 adr1-L1 adr1-L2 nrg1.1 nrg1.2* (*helperless*) plants were hand-infiltrated with *Pf0* carrying an EV or expressing AvrRps4, AvrRpm1, or AvrRpt2 (OD$_{600}$ = 0.2). Samples were collected before treatment and 0.5 hpi, 4 hpi, and 8 hpi. (A) Heatmap showing the normalized expression (z-score) of all the genes differentially regulated in at least one condition (FDR-adjusted $p < 0.05$, fold change > 2). Transcriptional response corresponding to PTI and all ETIs involve mostly the same genes. The impact of RNLs is clearly visible during RPS4 ETI. (B) Principal component analysis showing the effect of pathogen treatment on gene

expression in the different genotypes at time 0 (black circle), 0.5 hpi (green circle), 4 hpi (orange circle), and 8 hpi (red circle). The effects of ETI on gene expression are visible at 4 and 8 hpi but not at 0.5 hpi. Most of the variability observed is explained by time, then by treatment type, and lastly by genotype. (C) Venn diagrams comparing PTI-triggered gene up-regulation in Col-0, *adr1 triple*, *nrg1.1 nrg1.2*, and *helperless* mutants at 0.5 hpi, 4 hpi, and 8 hpi with *Pf0*-EV. PTI is largely RNL independent. (D) Venn diagrams comparing ETI-specific gene up-regulation in Col-0 and the *helperless* plants at 4 hpi and 8 hpi with *Pf0-AvrRps4*, *Pf0-AvrRpt2*, or *Pf0-AvrRpm1*. Notably, the vast majority of RPS4-induced gene expression is abolished in the *helperless* mutant at 4 and 8 hpi, whereas RPS2 or RPM1-induced ETIs are largely RNL independent. Underlying numerical data are provided in S1 Data. Col-0, Columbia-0; ETI, effector-triggered immunity; EV, empty vector; FDR,; hpi, hours post infection; NA, no application/treatment; OD$_{600}$, optical density at 600 nm; PC1, principal component 1; PC2, principal component 2; *Pf0*, *Pseudomonas fluorescens 0*; PTI, pattern-triggered immunity.

It was previously reported that SA accumulation during PTI is largely ADR1s dependent and that ADR1-L2 functions in SA-dependent and SA-independent feedback regulatory loops [14,47]. We therefore looked at the behavior of genes induced during *Pf0-1* EV infections (PTI-induced genes) during ETI (S3 Fig). We observed a very small quantitative effect of the *adr1 triple* and the *helperless* mutation on gene expression at 8 hpi with *Pf0-1* EV. We observed that PTI-induced genes were over-induced during *Pf0-1* AvrRps4 infections in an RNL-dependent manner and during *Pf0-1* AvrRpt2 and *Pf0-1* AvrRpm1 infections in an RNL-independent manner (S3 Fig). The NRG1s partially contributed to the induction of SA-related genes during RPS4/RRS1-mediated ETI (S4 Fig), as seen by the severely affected induction of gene expression at 8 hpi in the *helperless* mutant compared to *adr1 triple* (S5B Fig). Considering that ADR1s and NRG1s are involved in basal resistance against virulent pathogens and SA signaling, it would be interesting to observe the impact of RNLs on *Pf0-1* EV-induced (PTI-related) transcriptome regulation at later time points.

Together, our data demonstrate that *Pf0-1* EV-induced (PTI-induced) transcriptional reprogramming—at least up to 8 hpi—is to a great extent RNL independent. However, these PTI-induced genes are over-induced in an RNL-dependent manner during TNL-mediated ETI and in an RNL-independent manner in CNL-mediated ETI.

## RNL requirements in transcriptional reprogramming during ETI

We wanted to analyze whether the different requirements of the 2 RNL families in ETI mediated by the CNLs RPS2 and RPM1 and the TNL pair RPS4/RRS1 is also reflected in the transcriptional reprogramming. The effect of NLR activation on gene expression started to be visible at 4 hpi and was even more pronounced at 8 hpi (Fig 4B).

To better understand the impact of RNLs on ETI-specific gene regulation, we compared genes induced by the bacterial delivery of effectors to genes induced by the control *Pf0*-EV. The *helperless* mutant was incapable of mounting normal ETI after *Pst* DC3000 *AvrRps4* infiltration (Fig 1 and Fig 2). This is consistent with the result that, upon infiltration of *Pf0-1 AvrRps4*, 86.8% and 96.9% of the genes induced in Col-0 at 4 and 8 hpi, respectively, were not induced in the *helperless* mutant (Fig 4D, Table 1). Response to *Pf0-AvrRpt2* was also affected at 4 hpi in the *helperless* mutant with 47.2% of the genes induced in Col-0 requiring RNLs. However, the impact of the *helperless* mutant was strongly reduced at 8 hpi with *Pf0-AvrRpt2*, whereas only 25.3% of the control ETI response was affected. A similar tendency was observed with *Pf0-AvrRpm1* infections; 39% and 16.8% of the response was lost in the *helperless* mutant at 4 hpi and 8 hpi, respectively (Fig 4D, Table 1). Many genes were found to be differentially expressed specifically in the *helperless* mutant (Fig 4D).

In summary, RNLs contribute weakly to overall transcriptional reprogramming during PTI, but are fully required for TNL-triggered transcriptional reprogramming and partially required for CNL-mediated transcriptional reprogramming. This partial requirement was more pronounced at the early stage of CNL-mediated ETI. Further, RNLs function in the induction of SA-related gene expression during PTI and ETI responses, with a predominant role of the ADR1s.

**Table 1. Fractions of ADR1s and/or NRG1s synergistic, redundant, and specific gene up-regulation during ETI.**

| | RPS4 | | RPS2 | | RPM1 | |
|---|---|---|---|---|---|---|
| | **4 hpi** | **8 hpi** | **4 hpi** | **8 hpi** | **4 hpi** | **8 hpi** |
| **RNL dependent** | **86.8%** (486/560) | **96.9%** (1,404/1,449) | **47.2%** (417/883) | **25.3%** (463/1,831) | **39.0%** (667/1,712) | **16.8%** (458/2,725) |
| →*Shared regulation* | **67.1%** (326/486) | **60.0%** (842/1,404) | **59.5%** (248/417) | **63.3%** (293/463) | **67.0%** (447/667) | **65.3%** (229/458) |
| ➤ Synergistic | 63.6% (309/486) | 36.5% (512/1,404) | 46.0% (192/417) | 35.4% (164/463) | 52.3% (349/667) | 44.3% (203/458) |
| ➤ Redundant | 3.5% (17/486) | 23.5% (330/1,404) | 13.4% (56/417) | 27.9% (129/463) | 14.7% (98/667) | 21.0% (96/458) |
| → *Specific regulation* | **32.9%** (160/486) | **40.0%** (562/1,404) | **40.5%** (169/417) | **36.7%** (170/463) | **33.0%** (220/667) | **34.7%** (159/458) |
| ➤ ADR1s specific | 74.4% (119/160) | 72.4% (407/562) | 78.1% (132/169) | 60.6% (103/170) | 62.3% (137/220) | 49.7% (79/159) |
| ➤ NRG1s specific | 25.6% (41/160) | 27.6% (155/562) | 21.9% (37/169) | 39.4% (67/170) | 37.7% (83/220) | 50.3% (80/159) |

Fractions are defined according to the gene sets and functional categories defined in the text. The "RNL-dependent" fraction is the fraction of "ETI-regulated" genes that require RNLs, whereas the others are fractions of the "RNL-dependent" category. "Shared" is the addition of synergistic and redundant fractions, and "Specific" is the addition of "ADR1s-specific" and "NRG1s-specific" fractions. Fractions of main interest are in bold.

**Abbreviations:** ADR1, ACTIVATED DISEASE RESISTANCE 1; ETI, effector-triggered immunity; hpi, hours post infection; NRG1, N REQUIEREMNET GENE 1; RNL, RPW8 CC-domain NLR; RPM, RESISTANCE TO P. SYRINGAE PV MACULICOLA 1; RPS2, RESISTANT TO P. SYRINGAE 2; RPS5, RESISTANT TO P. SYRINGAE 5

## Sub-functionalization of ADR1s and NRG1s

We then sought to identify possible synergistic, redundant, and/or specific effects of the 2 RNL families on gene expression regulation during ETI. "ETI-regulated" gene sets were obtained by comparing *Pf0*-EV–induced genes to *Pf0*-effector–induced genes in Col-0 at 4 and 8 hpi. We first classified the genes into "RNL independent" and "RNL dependent" categories based on their behavior in the *helperless* mutant (S1 Table). We further divided those RNL-dependent genes into 4 functional categories: "ADR1 + NRG1 dependent" are the genes that are not differentially expressed in the absence of either ADR1s or NRG1s; "ADR1/NRG1 redundant" are "RNL dependent" genes that are still differentially expressed in either mutant (*adr1 triple* or *nrg1.1 nrg1.2*) compared to Col-0; and "ADR1 specific" genes are "RNL dependent" genes not expressed in *adr1 triple* mutants and still expressed in *nrg1.1 nrg1.2*. Inversely, "NRG1 specific" genes are "RNL dependent" genes not expressed in *nrg1.1 nrg1.2* and still expressed in *adr1 triple* (S1 Table).

We applied this categorization to differentially expressed genes (DEGs) during RPS4/RRS1-, RPS2-, and RPM1-mediated ETI. The lists of genes belonging to these categories can be found in S1 Dataset. We observed that gene up-regulation induced by RPS4/RRS1-mediated ETI is almost fully RNL dependent at 4 hpi (86.8%, Table 1) and at 8 hpi (96.9%, Table 1). During RPS4/RRS1-mediated ETI, 63.6% (309) of the "RNL dependent" genes required the combined action of both ADR1s and NRG1s at 4 hpi (Table 1). This fraction decreased to 36.5% (512) at 8 hpi (Table 1). However, the fraction of redundantly regulated genes increased from 3.5% (17) at 4 hpi to 23.5% (330) at 8 hpi. This shows that, at the onset of RPS4/RRS1-triggered ETI, both ADR1s and NRG1s are required, but at later stages one subfamily can partially substitute for the loss of the other. In total, 60% of the genes induced by RPS4/RRS1-triggered ETI were redundantly or synergistically regulated by ADR1s and NRG1s at 8 hpi, showing that helper function in gene regulation is mostly shared by both subfamilies.

A smaller fraction (32.9% and 40% at 4 and 8 hpi, respectively) of the "RNL dependent" gene regulation was found to be specific to either subfamily. This could be attributed largely to the ADR1s with 74.4% and 72.4% of the subfamily-specific gene expression regulation relying on ADR1s at 4 and 8 hpi, respectively (Table 1). A similar tendency was observed during infection with *Pf0*-AvrRpt2 (Table 1) or with *Pf0-1* AvrRpm1 (Table 1), especially at 4 hpi. In conclusion, our results show that, during all ETI responses tested, around 60% of RNL-dependent gene regulation is mediated by both ADR1s and NRG1s and that there is an important ADR1-specific gene regulation that cannot be compensated for by the NRG1s. These results are consistent with the unequal redundancy of ADR1s and NRG1s during basal defense and ETI. Further, our results suggest that RPM1 and RPS2 partially rely on the RNLs during the early steps of the transcriptional reprogramming.

## Classification of distinct NRG and ADR functions

To learn more about the nature of the RNL function, we looked at gene ontology (GO) terms associated with the "ADR1 + NRG1 dependent," "ADR1/NRG1 redundant," and "ADR1 or NRG1 specific" gene category during RPS4/RRS1-, RPS2-, and RPM1-mediated ETI responses at 4 and 8 hpi (S4 Fig, S2 Dataset). GO terms associated with Col-0 "ETI-regulated" genes were globally related to 4 categories: first, a large category grouping SA, systemic acquired resistance (SAR), JA, ROS metabolism, and defense; second, an HR category; third, a lipoprotein metabolism category; and finally, a category associated with ROS response, autophagy, and protein catabolic processes. In early ETI (at 4 hpi) the majority of up-regulated genes were involved in the category associated with SA/JA- and SAR-related pathways, ROS metabolism, and defense (S4 Fig). Later in ETI (at 8 hpi), up-regulated genes were also associated with the HR category; the lipoprotein metabolism category; and the category associated with ROS response, autophagy, and protein degradation. Our GO term analysis indicates that there was no GO term specifically associated with a particular sensor NLR-mediated ETI response. In other words, cellular processes transcriptionally (up-)regulated by the RNLs during RPS4/RRS1-mediated ETI did not differ strongly from the processes (up-)regulated by the 2 CNLs RPS2 (partially RNL dependent) or RPM1 (RNL independent). This observation prompted us to consider the possibility that RNLs—in RPS4/RRS1-mediated ETI—may ultimately regulate the expression of the same genes as RPM1 and RPS2, although the transcriptional regulation may differ quantitatively.

## RNLs act like CNLs

To investigate the hypothesis that RNLs and CNLs regulate the same genes, we looked at the normalized expression of ETI-regulated genes in Col-0 and the *helperless* mutant (Fig 5). The genes differentially regulated during ETI triggered by AvrRps4, AvrRpt2, or AvrRpm1 were largely overlapping, especially at 8 hpi (Fig 5A and 5C). Moreover, the expression of differentially regulated genes during RPS4-induced ETI, which reflects the action of RNLs, was also differentially regulated during RPS2- and RPM1-mediated ETI even in the absence of RNLs (Fig 5B and 5D, S6 Fig). Inversely, the expression of RPS2- and RPM1-regulated genes was globally sustained during RPS4/RRS1-mediated ETI through the action of RNLs, even though not all RPS2- and RPM1-regulated genes pass the threshold of statistical significance in RPS4/RRS1-mediated ETI (Fig 5B and 5D and S6 Fig). Consistent with this observation, the *RNL* deletion in *helperless* plants had a limited quantitative effect on the overall expression level of the RPS2- and RPM1-regulated genes during *Pf0-AvrRpm1* and *Pf0-AvrRpt2* infections (S6 Fig). Thus, RNLs support some quantitative effect on transcriptional regulation during RPS2-

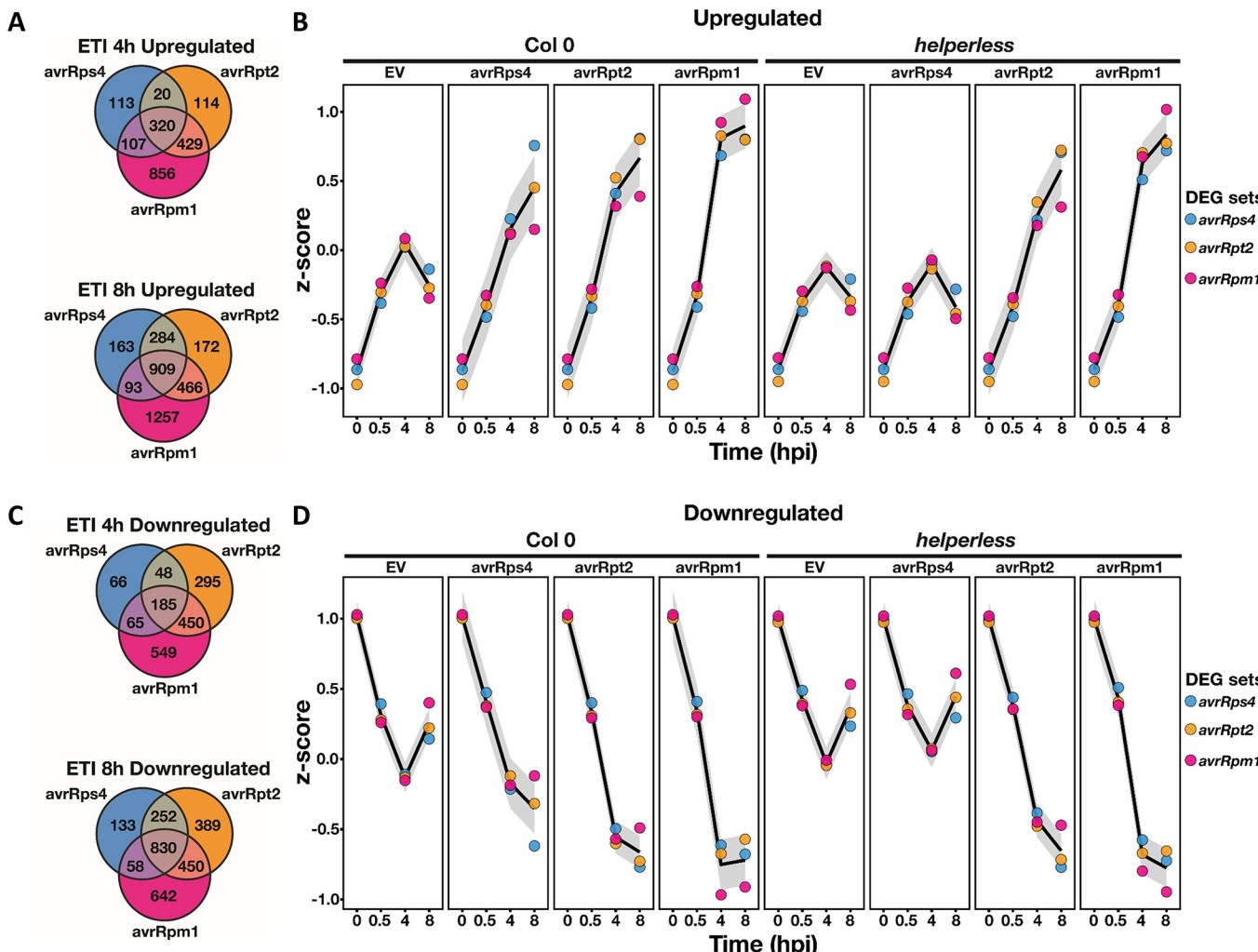

**Fig 5. RNLs function as classical CNLs.** Comparison of gene up-regulation (A, B) or down-regulation (C, D) across RPS4-, RPS2-, and RPM1-mediated ETIs. (A) and (C) Venn diagrams comparing up-regulated (B) or down-regulated (D) ETI-specific genes showing the extensive overlap between RPS4-, RPS2-, and RPM1-mediated ETIs. RPS4/RRS1 ETI, which reflects the action of RNLs, is very similar to CNL-mediated ETI. The curves in (B) and (D) show the normalized expression of the ETI-regulated gene sets, in Col-0 and *helperless* plants, across all conditions tested in the experiment. Notably, RPS4/RRS1-regulated genes (blue dots), which require RNLs during *Pf0-AvrRps4* infection, are differentially regulated by RPS2 and RPM1 in the absence of RNLs. Similarly, genes differentially regulated by RPM1 and RPS2 are also regulated by RNLs during *Pf0-AvrRps4* infections in Col-0, but the up- or down-regulation is weaker. Underlying numerical data are provided in S1 Data. CNL, coiled-coil domain-containing nucleotide-binding leucine-rich repeat receptor; Col-0, Columbia-0; DEG, differentially expressed gene; ETI, effector-triggered immunity; EV, empty vector; hpi, hours post infection; *Pf0*, *Pseudomonas fluorescens 0*; RNL, RPW8 CC domain containing NLR.

and RPM1-mediated ETI, but this effect is—at least in the case of RPM1—not required for an efficient immune response (S2A and S2C Fig).

These results demonstrate that effector-activated, TNL-induced, and RNL-mediated gene expression is very similar to that of an orthodox CNL-activated ETI, and that RNLs contribute quantitatively to CNL-mediated ETIs.

The apparent specificity of some genes for ADR1s may reflect the higher capacity of ADR1s to regulate gene expression. There was no strong enrichment of ADR1 or NRG1 family-dependent genes in any GO terms related to the ETI-mediated transcriptional reprograming. The "ADR1-specific" genes were in general not associated with a specific GO term distinct from the NRG1 family-regulated genes (S4 Fig). For example, we observed that genes related to SA

responses and SAR/SA pathways were strongly *ADR1* family-dependent at 4 hpi in all triggered ETI responses as well as in PTI (S5 Fig). This confirms that ADR1s function similarly or together with EDS1 and PAD4 in the regulation of the SA pathway at early stages of immunity and that this is most obvious in RPS4/RRS1-mediated ETI (S5 Fig, [27]). However, at 8 hpi with *Pf0-AvrRps4*, the up-regulation of the SAR/SA pathway genes in *adr1 triple* was similar to the 8 and 4 hpi samples of Col-0 and *nrg1.1 nrg1.2*, whereas these genes could not be activated in the *helperless* mutant (4 and 8 hpi). This shows that the *NRG1*s could compensate for the loss of the *ADR1*s at this time point (S4 Fig and S5 Fig). Overall, our results suggest that ADR1s and NRG1s redundantly and quantitatively regulate expression of the same genes during ETI and that the apparent specificity of some genes for ADR1s likely results from unequal redundancy. This also suggests that the specific function of the NRG1s in AvrRps4-triggered HR (Fig 2D) is independent of the ultimate transcriptional reprogramming and rather requires a specific function of NRG1s.

## Discussion

Recognition of pathogen-derived effectors by intracellular sensor NLRs triggers ETI, in many cases a strong immune response eventually leading to disease resistance and to HR. Many sensor NLRs mediate immune responses that require the presence of helper RNLs (S2 Table) [11–14,27]. Information on whether RNLs act redundantly or are required for specific immune pathways in sensor NLR-mediated ETI is still scarce. We provide a detailed, side-by-side comparison of immune responses in the *adr1 triple*, *nrg1.1 nrg1.2*, and *helperless* mutants to determine the specific, redundant, and synergistic functions of RNLs during immunity. This includes activation of effective disease resistance, HR initiation, and transcriptional reprogramming during PTI and ETI.

The *ADR1* RNL family was shown to be preferentially involved in defense activation and SA accumulation [11,14,26,27]. *A. thaliana* and *N. benthamiana NRG1*s, together with *EDS1* and *SAG101*, are required for TNL-mediated cell death and some TNL-dependent autoimmune phenotypes [11,13,27,48]. Thus, the current working model for RNL function and activity is that ADR1s mediate disease resistance and NRG1s are required for (at least TNL-triggered) cell-death signaling. Here, we observed that the apparent sub-functionalization of *ADR1*s and *NRG1*s is at least partly a case of unequal genetic redundancy. We and others described unequal redundancy in basal defense against *Psm* ES4326 and *Pst* DC3000 cor-, as well as in ETI mediated by RPS4/RRS1, RPP2, and RPS6 (Fig 1A and 1B, Fig 3B, [11]). Indeed, the loss of the 2 *NRG1* genes had no effect, whereas the *adr1 triple* was severely affected in the immune responses noted earlier. The *helperless* mutant was, however, as susceptible, or even more susceptible, than the respective sensor NLR mutant. This shows that the NRG1s can partially substitute for the loss of the ADR1 family function in mediating disease resistance but that the impact of the NRG1s on the overall phenotype is so small that it is only visible in the absence of the ADR1s. Inversely, the ADR1s can fully complement the loss of NRG1 function in resistance mediated by RRS1/RPS4, RPP2, and RPS6 sensor NLRs.

Unequal redundancy of RNLs is also revealed by the level of gene induction during ETI. If ADR1s and NRG1s were specialized to regulate certain genes or genetic pathways, we would expect that genes requiring NRG1s would be associated with a unique function/GO term that is not shared with ADR1s-requiring genes, and vice versa. We did not observe any such association. Genes induced by NRG1s alone or synergistically with ADR1s fall into the same functional categories as those induced by ADR1s (S4 Fig). On the contrary, we observed a quantitative effect of NRG1s and ADR1s on the overall gene expression during ETI (S6 Fig). In particular, ADR1s were shown to be specifically required for SA accumulation [12,14].

However, NRG1s affected the expression of SA metabolism and SA response genes, to a lesser extent, and this effect was only visible in the absence of *ADR1*s and most obvious in late stages of TNL-triggered ETI (S4 Fig, S5B Fig and S6 Fig).

In *A. thaliana*, it seems like *ADR1*s predominantly function to regulate/induce most ETI responses. In *N. benthamiana*, *NRG1* might play a more significant role, because it is required for cell-death induction downstream of all tested sensor TNLs or TIR domains (S2 Table, [12,13]). The requirements for either or both RNL families in NLR-mediated ETI might be different in other plant species and subjected to co-evolution with likely co-acting components such as EDS1, PAD4, or SAG101 [27,48]. Gantner and colleagues showed that ETI mediated by the *N. benthamiana* TNL Roq1 depends on the presence of *N. benthamiana EDS1* and *SAG101b* which signal with *NRG1*. In contrast, in *A. thaliana*, stable expression of *N. benthamiana Roq1* confers resistance to *Pst* DC3000 (naturally expressing the effector HopQ1 that is recognized by Roq1) only in the presence of *A. thaliana EDS1* and *PAD4*, which are associated with *ADR1* signaling [12,27,48]. This finding supports the speculation that, in *A. thaliana*, the *ADR1*s conserve most of the ancestral RNL function, whereas in *N. benthamiana*, *NRG1* seems to play a prominent role. Overall, our data and the current literature suggest that *ADR1s* and *NRG1s* are largely redundant gene families and that the function described in *A. thaliana* may vary in other species. It is still unclear whether the observed *N. benthamiana NRG1* specialization for TNL-mediated HR/cell death and resistance induction applies in *A. thaliana*, given the completely ADR1-dependent TNL RPP4-triggered resistance in *A. thaliana* Col-0 plants (Fig 2B, [14]).

During some ETI responses, ADR1s and NRG1s do not act redundantly but rather play specific roles. The induction of HR and defense by the TNL RPP4, the timely induction of HR upon *Pst* DC3000 *AvrRpt2* infection, and the bacterial growth restriction during *Pst* DC3000 *AvrPphB* and *Pst* DC3000 *AvrRpt2* specifically require ADR1s and not NRG1s (Fig 2A, 2B, 2C and 2E). Similarly, NRG1s are specifically required for HR after RPS4/RRS1 activation and for some autoimmune phenotypes (Fig 2D, [11,13,14,26,27]). Considering the fact that both ADR1s and NRG1s can regulate HR and defense independently (e.g., in RPP4- or RPP2-mediated ETI, auto-immune mutants, and sensor NLR or RNL activation-mimic mutants), sensor NLR specificity for either RNL family is likely to be the result of a preferential use of either ADR1s or NRG1s by the sensor NLR. This may include, but does not require, physical association with the relevant sensor NLR, as, e.g., shown for the tobacco sensor NLR N and its RNL helper NRG1 [22]. Specific and convincing interaction of the RNLs with sensor NLRs or with the transcriptional machinery involved in immunity is still lacking. Therefore, it will be of great value to gather detailed insights into RNL subcellular localization(s) before and after activation and to define RNL interactors regulating transcription.

A specific role of the ADR1s in SA signaling and SA accumulation is further supported by our *A. brassicicola* infections (Fig 3D and 3E), revealing a negative function of the ADR1s and EDS1 in the JA-dependent resistance against this necrotrophic fungal pathogen. The observed enhanced resistance against *A. brassicicola* in *eds1-12* could be explained by the reported function of EDS1 (together with PAD4) in inhibiting the transcriptional regulator MYC2, a basic helix-loop-helix leucine zipper motif containing transcriptional activator and master regulator of JA responses [49]. It is possible that the ADR1s also participate in this regulation. Thus, the ADR1s together with EDS1 and its partner PAD4 might contribute to the regulation of the interplay between SA and JA during plant immunity.

In *A. thaliana*, RPS2 and RPS5 ETIs are thus far the only analyzed contexts of CNL-mediated ETIs requiring RNLs for full resistance (this study; [11,13,14]). Although Castel and colleagues suggested a slight contribution of the *A. thaliana* NRG1s for RPM1-mediated HR at an early time point (4 hpi with *Pf0*-AvrRpm1; [13]), we were not able to confirm these findings in

our conditions (S2 Fig). Nevertheless, we observed that RNLs do participate in the regulation of gene expression during RPS2- and also RPM1-triggered ETI, in particular during the early stages of ETI (Table 1, Fig 5B and 5D, S6 Fig). However, RNLs were not required for RPM1-mediated defense or HR (S2A, S2C, S2F and S2G Fig). This suggests that RNLs support CNL-mediated ETI responses quantitatively, even though they do not always have a measurable impact on the defense phenotype.

RNL requirement for basal resistance was previously demonstrated for resistance against virulent *Pseudomonas* strains (*Pst* DC3000 and *Psm* ES4326) and the virulent *Hpa* isolate Emco5 [11,14]. We confirmed these results (Fig 3). However, while full basal resistance against *Psm* ES4326 requires the presence of both RNL families, basal resistance against *Pst* DC3000 seemed to only require *ADR1s* in our experimental conditions. This was indicated by the lack of a higher susceptibility of the *helperless* mutant compared to the *adr1 triple* (Fig 3A). We further demonstrated that, when the virulence function of the JA antagonist coronatine was removed (in infections with *Pst* DC3000 cor-), the *helperless* mutant was more susceptible than the *adr1 triple* mutant, resembling the findings for *Psm* infections [11]. This reveals that *NRG1s* function in basal resistance against *Pst* DC3000 like they do in resistance to *Psm* ES4326. However, this function is likely inhibited or antagonized by the effect of coronatine produced by *Pst* DC3000. Surprisingly, the coronatine-producing *Psm* ES4326 does not suppress NRG1s function, perhaps due to some unknown effector(s) contributing redundantly to this virulence function in *Pst* DC3000 [50,51]. We were unable to observe a function of RNLs during PTI (Fig 3C and S1 Fig). Therefore, our results and previously published findings suggest that RNLs are not required for signaling from at least the two RKs FLAGELLIN-SENSITIVE 2 (FLS2) and EF-TU RECEPTOR (EFR) [14]. The *Pseudomonas-A. thaliana* pathosystem that we used in our studies might not be the right tool to analyze RNL function in PTI, since resistance against virulent *Pseudomonas* strains strongly relies on RK function [4]. This idea is further supported by the lack of enhanced susceptibility in any RNL mutant during *Pst* DC3000 *ΔhrcC* infections compared to wild-type Col-0 (Fig 3C).

When we compared the ETI responses triggered by RPS2, RPM1, or RNLs (via RPS4/RRS1), we found that they involved similar genes (Fig 5A and 5C). Moreover, genes regulated by RNLs in RPS4/RRS1-triggered ETI were also regulated in RPS2- and RPM1-triggered ETIs, independently of RNL presence (Fig 5B and 5D, S6 Fig). Conversely, RNLs regulate the expression of RPS2- and RPM1-regulated genes during RPS4/RRS1-induced ETI (Fig 5 and S6 Fig). This shows that RNLs and CNLs regulate the same genes. Therefore, we suggest that RNL function is similar to CNL function in ETI. This hypothesis is further supported by the fact that, contrary to their strict requirement for TNL signaling, RNLs seem to act in parallel with CNLs. Indeed, RNLs are required for basal defense, which is a combination of RK, RLP, and weak NLR signaling, and thus RNL activation might not require or rely solely on sensor NLR activation. Most importantly, RNLs are not strictly required for any CNL function. For example, even though RPS2-triggered HR is delayed in *adr1 triple* and *helperless* mutants, RNLs are not required for RPS2 to ultimately induce HR (Fig 2E). Similarly, although RPS2 and RPM1 ETI-induced transcriptional reprogramming relies on RNLs at 4 hpi, RPS2 or RPM1 presence is sufficient for the regulation of most of the differentially regulated ETI genes at 8 hpi (Table 1, Fig 5C and 5D, S6 Fig). Overall, we propose that RNLs act like orthodox CNLs downstream of TNLs and support and enhance defense activation in parallel with CNLs. We note that RNLs are involved but not required for full defense activation during CNL ETI (Fig 6). Whether RNLs, representing a unique subclade of CNLs, also form oligomeric complexes upon activation, as do other CNLs [17,19,24], remains to be answered.

In summary, *A. thaliana* RNLs have 3 major functions in immunity (Fig 6). RNLs (1) bolster defense activation in the context of basal immunity (Fig 3, [14]), potentially as helpers for

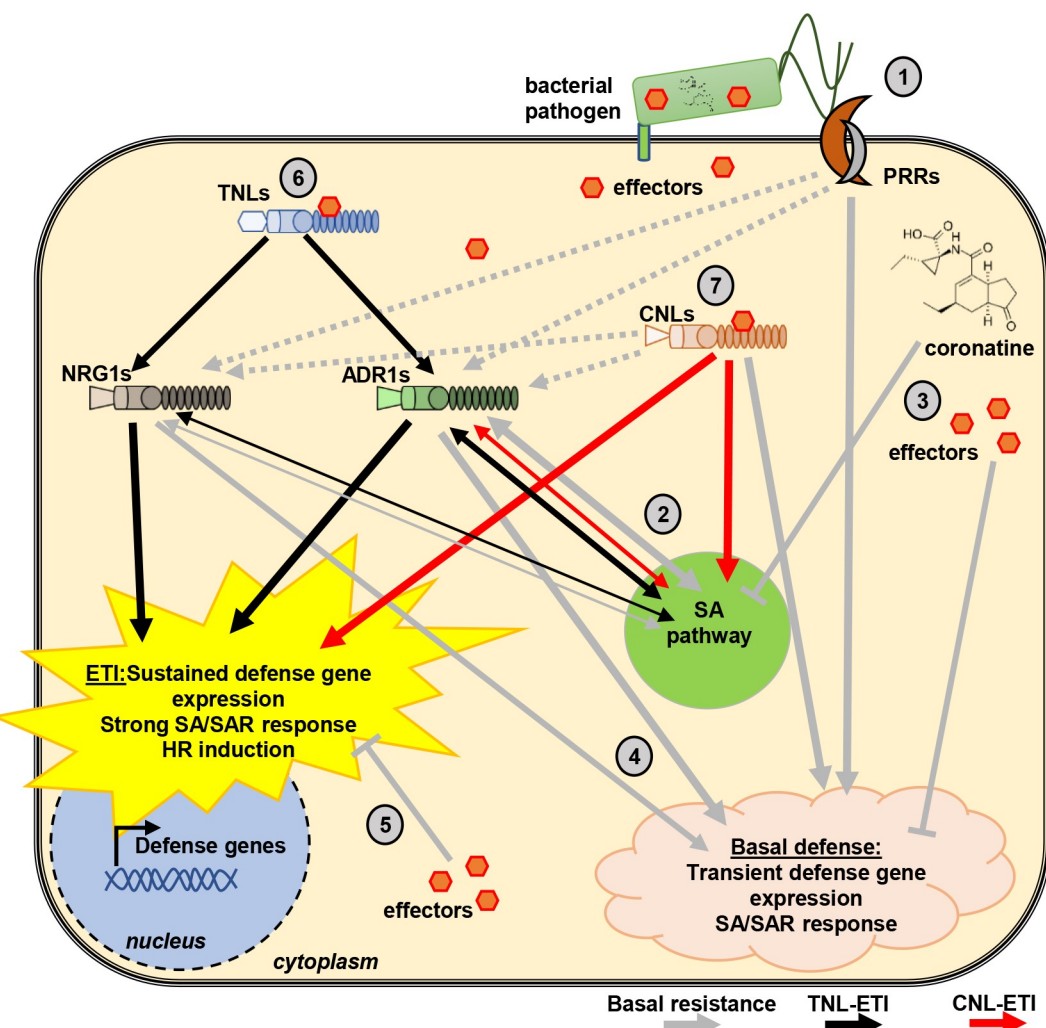

**Fig 6. Proposed model of RNL function in immunity.** Upon an infection of a plant cell by a pathogen, the first (early) response induced is PTI, irrespective of whether it is an avirulent or virulent pathogen. Thus, basal resistance and ETI happen in cells in which PTI signaling was already initiated and in some or the other way counteracted by effectors and other virulence molecules. Therefore, we propose that RNL function in immunity has to be considered as being part of a complex immune response network depicted in this proposed model. (1) Basal resistance (grey arrows) is initiated by the recognition of PAMPs by cell surface-localized PRRs. (2) PRR-triggered responses lead to the accumulation of SA and induction of SA responses, which requires RNLs [14]. (3) Pathogen-derived (virulence) effectors and the JA analog coronatine counteract PRR- and RNL-induced immunity and SA responses, thereby causing the so called (first) ETS. (4) Many pathogens, especially pathogenic bacteria, have effectors that can be recognized by some sensor TNL or CNLs and only induce a "weak remnant" ETI response [7] during basal resistance. This most likely leads to the activation of RNLs and can explain their requirement for basal resistance (see Fig 3). (5) It is possible that the aforementioned "weak" recognition and sensor NLR activation is also targeted by other effectors, causing the second ETS [62]. (6) RNLs are fully required for TNL-mediated immunity (black arrows) with some structural preferences for either ADR1s or NRG1s. After being activated by TNLs, RNLs act as CNLs to trigger strong and lasting defense activation as well as HR. ADR1s and NRG1s seem partially specialized in defense and HR, respectively, although the sub-functionalization is not strict. For example, ADR1s are specifically required for RPP4 signaling, while NRG1s are specifically required for RPS4-induced HR. (7) In addition, RNLs are involved but not required for CNL-triggered defense gene expression and HR (red arrows), further suggesting that RNLs act not directly downstream but in parallel with CNLs. If the sensor CNL is able to trigger a strong ETI by itself, the RNL involvement does not translate into requirement for proper disease resistance (e.g., RPM1 or ZAR1 ETI). Grey arrows indicate basal resistance (and PTI) signaling; black arrows indicate TNL-mediated ETI, and red arrows indicate CNL-mediated ETI signaling. Structural formula of coronatine was downloaded from Wikipedia (https://en.wikipedia.org/wiki/Coronatine). CNL, coiled-coil domain-containing nucleotide-binding leucine-rich repeat receptor; ETI, effector-triggered immunity; ETS, effector-triggered susceptibility; HR, hypersensitive response; JA, jasmonic acid; NLR, nucleotide-binding leucine-rich repeat receptor; PAMP, pathogen-associated molecular pattern; PRR, PAMP recognition receptor; PTI, pattern-triggered immunity; SA, salicylic acid; SAR, systemic acquired resistance; TNL, TIR domain-containing NLR.

weak ETI responses that fail to reach the threshold for HR; (2) mediate immune signaling from all TNLs tested so far, more or less redundantly (RPS6, RPS4/RRS1, RPP2, and RPP4 and TNL-dependent autoimmune phenotypes, Fig 1 and Fig 2, [11,14,26,27]); and (3) promote and contribute to timely defense activation and HR induction during CNL-mediated immunity (Fig 2E, Fig 5 and S6 Fig [14]); we further demonstrate that RNL function in TNL-mediated ETI resembles CNL function during sensor CNL-mediated ETI responses. Even though ADR1s and NRG1s are highly redundant in many aspects of their function in Arabidopsis, the ADR1s are more efficient in triggering defense-associated transcriptional reprograming than the NRG1s. Reciprocally, the NRG1s provide some specific functions, in particular the induction of HR during many TNL-mediated ETI responses.

## Materials and methods

### Plant material and growth conditions

All *A. thaliana* mutant lines are in the Col-0 background. The *adr1 triple* (*adr1-1 adr1-L1-1 adr1-L2-4* [14]), *nrg1.1 nrg1.2* [13], *rrs1a rrs1b* [30], *rpp2a* [52], *eds1-12* [53], *rps2-101C* [54], *rpp4* [55], *rpm1-3* [56], *zar1-3* [57], and *fls2* [58] mutants have been described previously. *zar1-3* seeds were kindly provided by Darrel Desveaux. The *eds1-12* mutant was kindly provided by Johannes Stuttmann and the *fls2* mutant by Georg Felix. *A. thaliana* plants were grown at short day conditions (8-hour light/16-hour dark cycle at 21˚C/18˚C and 45% humidity. *A. thaliana* sequence data for helper NLRs are available under the following AGI accession numbers: *ADR1/At1g33560*, *ADR1-L1/At4g33300*, *ADR1-L2/At5g04720*, and *NRG1.1/NRG1A/At5g66900*, *NRG1.2/NRG1B/At5g66910*.

### Generation of *helperless* mutant using CRISPR/Cas9

A CRISPR/Cas9 construct targeting *NRG1.1* and *NRG1.2*, previously used to generate a Col-0 *nrg1.1 nrg1.2* double mutant [13], was expressed in a Col-0 *adr1-triple* mutant [14]. Briefly, this construct contains a FAST-Red selectable marker, an intron-containing and plant-codon–optimized *Cas9* allele under the control of the *AtRPS5a* promoter and *E9* terminator, and an sgRNA targeting both *NRG1.1* and *NRG1.2* (GTGGAAAGCTGGTCTGAAG[nGG]) under the control of the *AtU6-26* promoter and terminator. In the first generation after transformation, we identified 3 lines (out of 16) with mutations in *NRG1.1* and in *NRG1.2*. By screening the non-transgenic T2 progenies of these lines, we identified a Cas9-free and *nrg1.1 nrg1.2* double mutant line. The mutations are a deletion of guanine 1153 and an insertion of adenine between bases 1161 and 1162 in the coding sequence (CDS) of *NRG1.1* and *NRG1.2*, respectively. These frameshift mutations lead to an early stop at amino acid position 398 and 394 in NRG1.1 and NRG1.2, respectively. This mutant line also contains T-DNA mutant alleles of *adr1*, *adr1-L1*, and *adr1-L2* [14]. We called the *adr1 adr1-L1 adr1-L2 nrg1.1 nrg1.2* pentuple mutant *helperless*.

### Bacterial infection assays

Plants for bacterial infiltration assays were grown for 6 weeks under short day conditions (8-hour light/16-hour dark at 21˚C/18˚C and 45% humidity). For bacterial growth curves, *Pst* DC3000 cor-, *Pst* DC3000 Δ*hrcC*, and *Pst* DC3000 expressing either *AvrRps4*, *AvrRpt2*, EV, *AvrRpm1*, *HopZ1a*, or *AvrPphB*—grown on KB plates containing appropriate antibiotics—were re-suspended in 10 mM $MgCl_2$ to a final concentration of $5 \times 10^5$ ($OD_{600}$ 0.001) or $1 \times 10^4$ colony-forming units (cfu)/mL ($OD_{600}$ 0.002) for the *Pst* DC3000 cor- and *Pst* DC3000 Δ*hrcC* strains. Plants were hand-infiltrated with the bacterial suspension. On the day of

infection (0 dpi), as well as at 3 dpi, leaf discs were taken and ground using a tissue lyser (Mill Retsch MM400, Retsch GmbH, Haan, Germany). Dilution series were plated on KB plates containing appropriate antibiotics, and cfu were counted after 2 days of growth at 28˚C. Statistical analysis from 4 replicates was done by one-way ANOVA Tukey's test using GraphPad Prism 8.0.2. Experiments were repeated at least 3 times with similar results. For the hypersensitive cell-death response (HR), *Pst* DC3000 expressing either *AvrRpt2*, *AvrRpm1*, *HopZ1a*, or *AvrPphB* were re-suspended in 10 mM $MgCl_2$ to $5 \times 10^7$ ($OD_{600}$ 0.1). *Pf0-1* (A*vrRps4*) was re-suspended to $1 \times 10^8$ ($OD_{600}$ 0.2). Bacterial solutions were hand-infiltrated into the right half of the leaves. Leaves were detached at the indicated time points, and autofluorescence was recorded by scanning the adaxial leaf sides using a Typhoon FLA9500 laser scanner (GE Healthcare, now Cytiva, Chicago, IL). Settings were as follows: Method: Alexa488; PMT: 450V; Laser: 473 nm. Image processing was done using ImageJ. For conductivity measurements, leaves were fully infiltrated with either *Pst* DC3000 expressing *AvrRpm1* ($OD_{600}$ 0.1) or *Pf0-1 AvrRpm1* ($OD_{600}$ 0.2). Leaf discs (5-mm diameter) were excised 1 hpi and incubated in water for 30 minutes. Five leaf discs were transferred together to the wells of a conductivity meter (CM100-2, Reid & Associates, Durban, South Africa) containing 3 mL distilled water, respectively. Conductivity of the solution was measured every hour for 24 hours. Statistical analysis from 5 replicates was done by one-way ANOVA Tukey's test using GraphPad Prism 8.0.2. The experiment was done twice with similar results.

## *A. candida* propagation and infection

For propagation of Ac2V, zoospores were collected from infected *A. thaliana* leaves, suspended in water (approximately $10^5$ spores/mL) and incubated on ice for 30 minutes. The spore suspension was then sprayed on plants using a Humbrol spray gun (approximately 700 μL/plant), and plants were incubated at 4˚C in the dark overnight. Infected plants were kept under 10-hour light (20˚C) and 14-hour dark (16˚C) cycles. Phenotypes were monitored 12 days after spraying.

## *Hpa* infections

*Hpa* isolates Cala2 and Emwa1 were propagated on the susceptible *A. thaliana* ecotypes Ler and Ws, respectively. Infections were conducted as described in [47]. Briefly, conidiospores were re-suspended in distilled water at a concentration of $5 \times 10^4$ spores/mL and used to spray-inoculate 10-day-old seedlings. Inoculated plants were covered with a lid to increase humidity, and sporangiophores were counted at 5 dpi.

## *A. brassicicola* infection assay

*A. brassicicola* MUCL 20297 cultivation and spore production was done as described earlier [59]. The *A. brassicicola* infection assay was conducted as described by [60]: *A. thaliana* plants were grown for 5 to 6 weeks under short day conditions (8-hour light/16-hour dark at 22˚C). *A. brassicicola* spores were diluted with sterile water to a final density of $1 \times 10^6$ spores/mL. Two leaves per plant were inoculated with two 5-μl droplets of the spore solution, respectively. Nine plants per genotype were used for each experiment. The experiment was done twice with similar outcomes. Infected plants were kept under 100% humidity. Fungal growth was monitored at 7, 10, and 13 dpi. Disease symptoms were classified into the following categories: 1 (no symptoms), 2 (light brown spots at infection site), 3 (dark brown spots at infection site), 4 (spreading necrosis), 5 (leaf maceration), and 6 (sporulation of the fungus). A disease index (DI) was calculated with the following formula: DI = $\Sigma$ i * $n_i$—"i" is the symptom category, and "$n_i$" is the percentage of leaves in "i." Symptom scores were statistically analyzed with a

two-way ANOVA and Tukey's test using GraphPad Prism 8.0.2. Pictures of representative infected leaves were taken 13 dpi with a Canon EOS 80D Body Camera. *A. brassicicola* spores were kindly provided by Birgit Kemmerling.

## Oxidative burst measurements

Leaf discs (4-mm diameter) were excised from leaves of 6-week-old plants and incubated in water overnight. Leaf discs ($n = 6$) were placed in a 96-well plate (1 disc/well, Greiner, Kremsmünster, Austria) containing 18 µg/mL luminol L-012 (Wako, Osaka, Japan) and 18 µM horseradish-peroxidase (AppliChem, Darmstadt, Germany). flg22 (Biomatik, Kitchener, Canada) was added to yield indicated concentrations. Luminescence was recorded over time using a TriStar$^2$ S LB 942 plate reader (Berthold Technologies, Bad Wildbad, Germany). flg22 peptide was kindly provided by Georg Felix. Peptide was dissolved in water and diluted in a solution containing 0.1% BSA and 0.1 M NaCl.

## RNA extraction

RNA was purified from plant tissue using the RNeasy 96 Kit (Qiagen, Hilden, Germany). The plant "user-developed protocol" was followed except for the addition of a 96% ethanol wash step after the second RPE wash to ensure removal of residual salts. Purified RNAs were kept at −80°C.

## RNA sequencing

Illumina-based mRNA sequencing libraries were prepared from 1 µg RNA following Finkel and colleagues [61]. Briefly, mRNAs were selected using Sera-mag oligo(dT) magnetic beads (GE Healthcare Life Sciences). RNAs were washed and fragmented at 94°C for 6 minutes. First-strand cDNA synthesis was performed using random hexamers and reverse transcriptase (Superscript III reverse transcriptase, Invitrogen, Carlsbad, CA). Second-strand cDNA synthesis was done using DNA Polymerase I and RNAseH. Double-stranded cDNAs were end-repaired using T4 DNA polymerase, T4 polynucleotide kinase, and Klenow polymerase. The DNA fragments were then adenylated using Klenow exo-polymerase to allow the ligation of Illumina adapters (Kapa Dual-indexed adapter kit, Roche, Basel, Switzerland). Unless specified, reagents were purchased from Enzymatics. Library quality control and quantification were performed using the 5200 Fragment Analyser and the NGS fragment kit (Agilent Technologies, Santa Clara, CA). Libraries were sequenced using Illumina HiSeq4000 sequencers to generate 50-bp single-end reads.

## RNA sequencing read processing

Initial quality assessment of the Illumina RNA sequencing reads was performed using FastQC version 0.11.7. Trimmomatic version 0.36 was used to identify and discard reads containing the Illumina adaptor sequence. The resulting high-quality reads were then mapped against the TAIR10 Arabidopsis reference genome using HISAT2 version 2.1.0 with default parameters. The featureCounts function from the Subread package was then used to count reads that mapped to each one of the 27,206 nuclear protein-coding genes, and we used these counts to construct a raw count table of expression.

We used the package DESeq2 version 1.22.1 to define DEGs using the raw count table described earlier. For visualization purposes, we applied a variance stabilizing transformation to the raw count gene matrix. We then standardized (z-score) each gene along the samples measured. We utilized the package clusterProfiler to map the list of DEGs to gene ontologies;

we took the top 25 GO categories per set of DEGs to construct the heatmap of GOs. We utilized the function UpSet from the package ComplexHeatmap to construct the any upset plot shown along the manuscript. All necessary data and scripts to reproduce every part of the RNA sequencing analysis are deposited in https://github.com/isaisg/helperless.

## Supporting information

**S1 Fig. RNLs do not function in flg22-induced ROS burst.** Left panels: Oxidative burst in leaves of the indicated genotypes after addition of 1 nM, 10 nM, or 100 nM flg22. Results are means × SD ($n = 6$). Right panels: Total ROS production over 34 minutes after flg22 treatment. Values are means × SD ($n = 6$). Letters indicate statistically significant differences following ANOVA with Tukey's test ($\alpha = 0.05$). Experiment was done 3 times. Underlying numerical data are provided in S1 Data.
(TIF)

**S2 Fig. RNLs are not required for RPM1, ZAR1, and RPS5 (CNL)-mediated ETI responses.**
(A, B) Six-week-old plants were hand-infiltrated with *Pst* DC3000 (A), *AvrRpm1* ($OD_{600} = 0.001$), or (B) *HopZ1a* ($OD_{600} = 0.001$), and bacterial growth was assessed at 0 and 3 dpi. Box limit represents upper and lower quartile; maximum and minimum values are displayed in whiskers. The middle line shows the median, the cross the mean cfu/cm². Dots represent 4 technical replicates (leaf discs) in one experiment (biological replicate). Experiment was done 3 times with similar results. Letters indicate statistically significant differences following ANOVA with Tukey's test ($\alpha = 0.05$). (C–E) The right leaf half of 6-week-old plants was hand-infiltrated with (C) *Pst* DC3000 *AvrRpm1* ($OD_{600} = 0.1$), (D) *Pst* DC3000 *HopZ1a* ($OD_{600} = 0.1$) or (E) *Pst* DC3000 *AvrPphB* ($OD_{600} = 0.1$). A Typhoon laser scanner was used to detect autofluorescence of dead leaf tissue at indicated time points. Representative leaves shown in a false color scale (black to blue: healthy leaf tissue, orange to white: dead). NLRs activated in infection experiments shown in A–E are indicated in parenthesis. (F, G) Leaves of 6-week-old *A. thaliana* plants were fully hand-infiltrated with either (F) *Pst* DC3000 expressing *AvrRpm1* ($OD_{600} = 0.1$) or (G) *Pf0-1* expressing *AvrRpm1* ($OD_{600} = 0.2$). Twenty-five leaf discs were collected and rinsed in deionized water, and conductivity of 5 leaf discs immersed in 3 mL deionized water was measured at 4, 8, 12, and 16 hpi. Values are means of conductivity [μS/cm²] ($n = 5$). Letters indicate statistically significant differences following ANOVA with Tukey's test ($\alpha = 0.05$). Experiment was done twice with similar results. Underlying numerical data are provided in S1 Data.
(TIF)

**S3 Fig. RNLs and CNLs enhance the expression of PTI-regulated genes during ETI.** Expression profile of PTI-regulated genes at 0.5 hpi, 4 hpi, and 8 hpi with *Pf0-1* EV, *Pf0-1 AvrRps4*, *Pf0-1 AvrRpt2*, or *Pf0-1 AvrRpm1*. PTI genes, which are induced by *Pf0-1* EV infections, are over-induced during RPS4 ETI in an RNL-dependent manner. Underlying numerical data are provided in S1 Data.
(TIF)

**S4 Fig. GO terms associated to the RNL function.** GO categorization of ETI-up-regulated genes. "Col-0 ETI" refers to the "ETI-regulated" gene set of Col-0. Activation of RPS4/RRS1, RPS2, and RPM1 induce genes involved in the same processes. There is no category associated to "ADR1 specific" that is distinct from the ones regulated redundantly or synergistically by NRG1s and ADR1s. For details see S2 Dataset. Underlying numerical data are provided in S1 Data.
(TIF)

**S5 Fig. Effect of loss of RNLs on transcriptional reprogramming of SA-related genes during *A. thaliana* immunity.** Comparison of SA-related gene expression in *Pf0-1* EV (A), *Pf0-1 AvrRps4* (B), *Pf0-1 AvrRpt2* (C), and *Pf0-1 AvrRpm1* (D) infiltrations at 0.5 hpi, 4 hpi, and 8 hpi. Genes are clustered according to their expression changes throughout the different samples and time points. Highest (red) and lowest (blue) log2 fold change is shown in heatmap. Visualization and clustering done with CLC Main workbench 20 (QIAGEN Aarhus A/S; www.qiagenbioinformatics.com). Underlying numerical data are provided in S1 Data.
(TIF)

**S6 Fig. The effect of *RNL* mutants on ETI-regulated gene expression is quantitative.** Normalized expression level of RPS4/RNLs up- (A) or down- (B) regulated genes at 4 and 8 hpi in Col-0 (circles), *adr1 triple* mutant (squares), *nrg1.1 nrg1.2* (diamonds), and *helperless* (triangles) mutants, during *Pf0-1* EV, *Pf0-1* AvrRps4, *Pf0-1* AvrRpt2, or *Pf0-1* AvrRpm1 infection. Colors indicate a statistical difference (post hoc ANOVA, adjusted $p < 0.05$). Notably, RNL loss-of-function mutants affect gene expression quantitatively. This effect is most striking at 8 hpi during RPS4-mediated ETI but is also visible during RPS2- and RPM1-mediated ETI. Underlying numerical data are provided in S1 Data.
(TIF)

**S1 Table. Expected expression profiles of ETI-induced genes according to their RNL requirement.**
(DOCX)

**S2 Table. Overview of helper RNL requirements in (auto)immunity.**
(DOCX)

**S1 Dataset. Up- and down-regulated helper NLR-dependent genes.**
(XLSX)

**S2 Dataset. GO terms associated to Col-0 ETI-induced genes and RNL-dependent ETI-induced genes during Pf0-1 AvrRpt2, AvrRps4, or AvrRpm1 infection.**
(XLSX)

**S1 Data. Summary of all numerical data presented in Fig 1, Fig 2, Fig 3, Fig 4, Fig 5 and S1 Fig, S2 Fig, S3 Fig, S5 Fig and S6 Fig.**
(XLSB)

**S1 RNAseq Data. Full gene expression data of all tested genotypes, time points and treatments.**
(XLSB)

## Acknowledgments

We thank Christel Kulibaba-Mattern and Sarina Schulze for technical support. We would also like to thank Dr. Marc Nishimura for valuable discussions and critical reading of the manuscript and all El Kasmi, Dangl, and Jones lab members for helpful comments and discussions.

## Author Contributions

**Conceptualization:** Svenja C. Saile, Pierre Jacob, Jonathan D. G. Jones, Jeffery L. Dangl, Farid El Kasmi.

**Data curation:** Svenja C. Saile, Pierre Jacob, Baptiste Castel, Lance M. Jubic, Isai Salas-Gonzáles.

**Formal analysis:** Svenja C. Saile, Pierre Jacob, Baptiste Castel, Lance M. Jubic, Isai Salas-Gonzáles, Marcel Bäcker, Farid El Kasmi.

**Funding acquisition:** Jonathan D. G. Jones, Jeffery L. Dangl, Farid El Kasmi.

**Investigation:** Svenja C. Saile, Pierre Jacob, Baptiste Castel, Lance M. Jubic.

**Methodology:** Svenja C. Saile, Pierre Jacob, Baptiste Castel, Lance M. Jubic, Isai Salas-Gonzáles, Marcel Bäcker.

**Project administration:** Jonathan D. G. Jones, Jeffery L. Dangl, Farid El Kasmi.

**Resources:** Jonathan D. G. Jones, Jeffery L. Dangl, Farid El Kasmi.

**Supervision:** Jonathan D. G. Jones, Jeffery L. Dangl, Farid El Kasmi.

**Validation:** Svenja C. Saile, Pierre Jacob, Baptiste Castel, Isai Salas-Gonzáles, Jonathan D. G. Jones, Jeffery L. Dangl, Farid El Kasmi.

**Visualization:** Svenja C. Saile, Pierre Jacob, Baptiste Castel, Isai Salas-Gonzáles.

**Writing – original draft:** Svenja C. Saile, Pierre Jacob, Farid El Kasmi.

**Writing – review & editing:** Svenja C. Saile, Pierre Jacob, Baptiste Castel, Lance M. Jubic, Isai Salas-Gonzáles, Jonathan D. G. Jones, Jeffery L. Dangl, Farid El Kasmi.

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
