## [Editor Report · Decision Letter 0]

28 Feb 2020

Dear Dr El Kasmi, 

Thank you for submitting your manuscript entitled "Arabidopsis defense activation by sensor NLRs requires two unequally redundant helper NLR families acting as classical CNLs" for consideration as a Research Article by PLOS Biology.

Your manuscript has now been evaluated by the PLOS Biology editorial staff as well as by an academic editor with relevant expertise and I am writing to let you know that we would like to send your submission out for external peer review.

Please re-submit your manuscript within two working days, i.e. by Mar 01 2020 11:59PM.

Kind regards,

Di Jiang

PLOS Biology

---

## [Decision Letter · Decision Letter 1]

2 Apr 2020

Dear Dr El Kasmi,

Thank you very much for submitting your manuscript "Arabidopsis defense activation by sensor NLRs requires two unequally redundant helper NLR families acting as classical CNLs" for consideration as a Research Article at PLOS Biology. Your manuscript has been evaluated by the PLOS Biology editors, an Academic Editor with relevant expertise, and by four independent reviewers.

In light of the reviews (below), we will not be able to accept the current version of the manuscript, but we would welcome re-submission of a much-revised version that takes into account the reviewers' comments. In addition to addressing the reviewers' concerns, you will need to edit the manuscript thoroughly to increase its readability. We cannot make any decision about publication until we have seen the revised manuscript and your response to the reviewers' comments. Your revised manuscript is also likely to be sent for further evaluation by the reviewers.

We expect to receive your revised manuscript within 2 months. 

**IMPORTANT - SUBMITTING YOUR REVISION**

*Re-submission Checklist*

*Published Peer Review*

*PLOS Data Policy*

*Blot and Gel Data Policy*

Sincerely,

Di Jiang, PhD

Associate Editor

PLOS Biology

REVIEWS:

Reviewer #1: The manuscript by Saile, Jacob et al. deals with the functional characterization of helper NLRs belonging to the RNL class in Arabidopsis thaliana. There are two different families of RNLs in Arabidopsis, the ADR1s and NRG1s. The three ADR1s are dispersed in the genome, and triple mutant plants were previously analyzed in detail (e.g. Bonardi 2011). However, analysis of NRG1s was previously hampered by tight genetic linkage. With increased availability of tools for targeted mutagenesis, several groups generated nrg1 mutant lines in N. benthamiana (Qi et al. PNAS, Castel et al. NPH) and/or Arabidopsis (Wu et al. NPH, Castel et al. NPH, Lapin et al. TPC), and also lines lacking both NRG1s and ADR1s were generated (Wu et al, Lapin et al). Analysis of these mutant lines revealed that

i) NRG1 appears essential for all immune responses mediated by TNLs (but not CNLs) in N. benthamiana (Qi et al, Wu et al)

ii) NRG1s are also important for a number of TNL-mediated resistance responses in Arabidopsis (e.g. RPS4/RRS1, RPP1), but some TNLs also signal via ADR1s. Resistance mediated by CNLs is not (RPS5, MLA) or only slightly (RPM1, RPS2) impaired (Castel et al)

iii) Different TNLs show various degrees of dependence on either ADR1s or NRG1s (mainly tested with autoimmune outputs), and ADR1s and NRG1s appear to function independently or synergistically in "basal resistance" (Psm) or TNL-mediated (RPS4, RPS6) resistance (Wu et al)

iv) One intriguing finding was that, in RPS4/RRS1-mediated resistance, ADR1s are more important to limit bacterial growth than NRG1s, but cell death was strongly reduced in nrg1 mutant, but not adr1 mutant plants (Castel et al, Lapin et al). This observation sparked the hypothesis that NRG1s are required for cell death and ADR1s for resistance signaling, implicating also that these are two independent processes.

The current manuscript provides a systematic and detailed comparison of Arabidopsis adr1 and nrg1 mutant lines with a "helperless" quintuple mutant line. In a first section of the manuscript, mutant lines are used in infection and HR assays to probe the requirement of a number of immune receptors for the helpers. Presented data confirms and extends previously reported findings, and reveals important contributions of NRG1s to several immune responses that only become visible in absence of ADR1s in the helperless line. In contrast to specific functions of NRG1s in cell death and ADR1s in resistance proposed before, the authors favor a model of genetic redundancy and synergism, with a certain degree of specialization visible in some comparisons. 

In a second section of the manuscript, different mutant lines are used for rather large scale transcriptomics (312 samples?). This section globally supports findings of immunity assays etc, and further corroborates a minor contribution of hNLRs also to CNL-mediated resistance (RPM1, RPS2). An important feature of the underlying experiments is that modified Pseudomonas fluorescens strains were used - reducing the variables indeed to the one effector that is translocated (or not) by the bacteria in individual treatments. This section provides important information and conclusions, e.g. a lacking specialization of NRG1 or ADR1 regulons (arguing against functional specialization of the hNLRs) or overall resemblance of the hNLR-regulated transcriptome to a "classical" CNL transcriptome. It is my impression that this 2nd section of the manuscript requires substantial revision, in which the authors should improve the structure and also apply reductionism to guide readers towards the essential. I also feel that explanations in the main text, figure legends and methods for this section require improvement, but must admit that I am myself not a specialist for transcriptomics - other reviewer(s) may be more helpful here.

Overall, the manuscript may be considered as lacking novelty, as large parts confirm recent reports on helper functions. At the same time, the fine-grained transcriptome analyses adds to previous data, and the authors provide an alternative hypothesis on functions of hNLRs - not specialized for resistance (ADRs) and cell death (NRGs), but rather redundant and/or synergistic, with some sensor NLRs preferentially using one or the other helper class. It is important to actually provide this alternative hypothesis, as whether or not HR and resistance belong to the same "program" or not is a notorious subject. Thus, it definitively adds new aspects.

Major points:

- The 2nd section of the manuscript is extremely difficult to read and would benefit from more reflected sub-structuring, as a (crude, un-reflected) suggestion i) presentation of the transcriptomics study in general, ii) hNLRs and PTI, iii) hNLRs and ETI, iv) specialization etc

- Supplemental figures are not cited in order (why?). 

- Supplemental figure S2 is not referred to at all

- The simple amount of supplemental data appears overwhelming, and a large portion of the results presents mainly data included in supplemental figures.

- Supplemental figure S3 is used to illustrate expression profiles underlying categorization for further analyses. However the graphs shown in this figure are composed of made-up data entered to Excel (if I get things right), the y-axes do not have a unit, and there are even error bars (derived from?). This suggests that "true" data is at the basis of the graphs, which is inappropriate. The same message could be illustrated in form of a table or a (flow) scheme. 

- line 304: what is meant with Table 1 here? Not that simple/obvious to come from the Venn diagrams to the percentages mentioned in the same sentence.

- ~ line 307: "16.8 % of the response was lost in helperless (5E, 5F). It is the same type of info as previously (lines 302-304, where figure 4D was cited), but now figures 5E,F are cited? And again Table 1? There are also 873 genes that are upregulated (in this comparison) in helperless, but not Col. Which is even more than those genes upregulated in Col, but not in helperless. Any comment on that?

- line 333 and following: Again, what is Table 1?

- Figure 5: The bar graphs only represent the numbers directly printed above. At the same time, the text only cites percentages, which can be calculated from these numbers. This is confucing, and I do not see any point for displaying these data as graphs. A table could display these data more efficiently, and would give the simple opportunity to both indicate absolute numbers and percentages - which are mentioned in the text (lines 333 and following). I propose to convert data into a table to eliminate Figure 5 (again, y-axes go without units). Also, the entire paragraph about RNL-specific regulons is lengthy

- lines 378 and following: "RNLs regulate the same genes as CNLs (S4 Fig)". Looking at the figure S4 and also considering colors (q value), this is not obvious. It is visible that the counts (the size of the circles) are similar… but still. The legend (methods) should include some comment on q-value, and I am wondering whether it might not be possible to display this type of data in a different (simpler) manner to make the conclusion more obvious to extract, more accessible.

- Figure 6, legend: (A,B) is correct, (B,D) should say (C,D)

- One important conclusion of the transcriptome analysis is that expression changes mediated by RNLs (and thus by TNLs as sensors, in the case of RPS4/RRS1) are similar to those of classical CNLs. Reports on structural elucidation of the resistosome from 2019 (Wang et al, Science) put these findings to some extent into context, as further reports suggest that the resistosome mechanism might be one major mechanism for signaling by different CNLs (eg Hadachi et al., eLife 2019; Li, Habring et al. Cell Host Microbe 2020). These reports should be discussed and also be included in intro. 

- Similarly, a recent report on a Pst effector compendium (Psy-Tec; Laflamme et al. Science 2020) showed an interesting positive correlation between the strength of resistance and the occurrence of macroscopic HR - further corroborating the notion that these are not distinct pathways as concluded from the transcriptomics. This should also be included in the discussion.

- line 464: the qi 2018 PNAS reference should also be listed here, as NRG1 was not analyzed in the mentioned reference

- line 522 and following: It is stated in the article that "basal resistance" also includes weak ETI (lines 209-10). Thus, the statement made here is contradictory. This is actually taken up again in line 537, but now with a different statement.

- line 551 and following: Nicotiana benthamiana lines are mentioned in the methods, but were not used in experiments, if I am not mistaken

minor points:

- line 58 "…to dampen PTI [5]". This may not be the most appropriate citation here, as the cited review focusses rather on effector functions different from PTI suppression.

- lines 64 - 65: Consider re-phrasing. The sentence may imply that there are 3 classes of NLRs, and these are further divided in sub-classes.

- lines 78 - 80: The statement about the used nomenclature is a bit confusing, as no alternative nomenclature is mentioned. It might be helpful to refer to the alternative NRG1A etc names used e.g. in Wu et al.

- line 104: Consider rephrasing. It is rather the (immune) phenotype of helperless mutant plants that was examined than the helperless phenotype.

- line 132: Shouldn't it say "different requirements for"?

- line 138: cooperative function of the TNLs

- lines 150 - 152: Results referred to in this phrase do not indicate recognition of AvrRps4 by another TNL (pair), as a direct comparison with DC3000 would be required for this statement. Also, experiments shown in Saucet 2015 (Nat Comms) suggest that this is not the case (equal susceptibility of rrs1a rrs1b to AvrRps4 and DC3000). This suggests rather that additional (weak) recognition events may be impaired in the helperless line

- line 154: [10, 12] Citation [10] is inappropriate for this statement, as an rrs1a rrs1b line was not used in this study. The Saucet 2015 reference should be used here.

- line 198: I think that journal policy requires figure panels to be referred to in order. Here, E is cited before D.

- line 217 …helperless plants

- line 232 and following: figure 3 (or rather data therein) cannot demonstrate anything in respect to Hpa, as only Pst was used in the experiments. The comment on a "slight contribution" is confusing, as no effect is stated a few lines before. Also, this is not supported by statistics and not convincing in the figure.

- line 274: 4B cited before 4A

- line 283: infectionsin

- lines 287 and following: The statement about the Pf0-EV transcriptome at later time points should be incorporated in the discussion

- line 294 and following: this clustering is really difficult to see and extract from the PCA. 

- line 310: "contribute weakly to basal resistance". Shouldn't it rather say PTI, when using Pfluorescens?

- ROS response should be spelled in capitals in S4

- line 390: "the expression … could be efficiently induced/repressed". Rephrase.

- line 604: infections

figure legends:

- Figure 1B: Statement about means not clear; consider rephrasing. 

- Figure 1C: "Activated NLRs are in parenthesis." is a bit confusing. Maybe add something like NLRs activated in infection experiments shown in A-C are indicated in parenthesis.

- figure s1: the " x 3 ml" in the unit definition appears a bit odd

- many figure legends could be more detailed

Reviewer #2: NLR proteins are central to plant immunity and consequently are of enormous academic and agricultural importance. 

The RNL subclass of NLR proteins are emerging as key components for the function of numerous NLRs.

However, despite their importance, there is currently only limited information regarding RNL activities associated with plant immune function. 

This manuscript provides deep new insights into RNL function associated with different branches of plant immunity. 

It is therefore both important and timely. 

The manuscript is both written and presented to a high standard. It is a beautiful paper full of important information that will be of great interest to the community. 

Consequently, I would strongly recommend this paper is accepted post a minor revision. 

The first link of the RNL class of proteins to plant disease resistance was by Grant et al. 2003.

It would therefore be helpful for readers if this paper was cited. 

Reviewer #3: The manuscript by Saile et.al reported an unequally redundant role of two helper NLR families ADR1s and NRG1s in Arabidopsis immune response. Plant NLR proteins functions as intracellular immune receptors that perceive pathogen effectors to induce ETI. Recently, two RPW8-like CC-NLR (RNL) families ADR1s and NRG1s were found serve as helper NLRs for sensor NLRs, especially TIR-NLRs (TNLs). However, the specificity and redundancy of these two helper NLRs remain less understood. By testing various pathogens and CNL-, TNL-typed effectors in adr1 triple, nrg1 double and helperless mutants, this manuscript reveals an unequally redundant role of ADR1s and NRG1s. 

Although the major conclusion of this study that ADR1s and NRG1s are required downstream of all sensor TNLs to activate ETI, and that both RNL families are required to support the activation of some CNLs-initiated ETI, was established by Wu et al. and Lapin et al. before, the authors tested more different effectors to present specificity of ADR1s and NRG1s in different TNL and CNL signaling pathways. The time-resolved genome expression profiling data also give more information of helper NLRs' function in transcriptional reprogramming during basal resistance and TNL-, CNL-mediated immunity. 

Minor:

No error bar and statistical analysis were shown in figure 1B and figure 2B. The authors also should indicate how many biological replicates were performed. 

The eds1 mutants were tested in figure 1B, figure 2B and figure 3A. However, eds1 mutant results were not shown in figure 1A, figure 2A C D E, and figure 3B. It is important to compare all data equally. 

The authors showed that helper NLRs ADR1s and NRG1s unequally and redundantly contribute for basal resistance. However, the helper NLRs do not function in flg22-induced ROS production. It seems further discussion is needed on how PTI is linked to helper NLRs.

It is a little confused by the colors of arrows in figure 7. Are the pathways connected by the red arrow and the pathways connected by the black arrow are independent or integrated? The authors showed that the helper NLRs are involved in some of the CNL-mediated immunity. Why there is no arrow indicating connection between helper NLRs and CNLs?

Reviewer #4: In this manuscript, the authors further characterize the role of ADR1s and NRG1s in TNL- and CNL-mediated ETI responses and also in basal resistance. 

In agreement with the previously published report (Ref #10), the data presented in this paper also show that ADR1s and NRG1s function in an unequally redundant manner in RPS4/RRS1- and RPP2-mediated resistance. Furthermore, the data presented indicate that ADR1s and NRG1s function redundantly in TNL WRR4-meidated resistance.

Consistent with the previous report (Ref #10), NRG1s are required for RPS4-mediated HR and ADR1s are required for RPS2-mediated HR and disease resistance. In addition, the data presented also shows that ADR1s play a role in basal defense. Here, authors use Pst DC3000 cor- strain to uncover the role of NRG1 in basal defense.

The data presented clearly shows that ADR1s and NRG1s have no role in RPM1-, RPS5- and ZAR1-mediated ETI responses. 

Most importantly in this paper, the authors present a significant amount of time-resolved transcriptome data to determine the role of RNL in shaping the transcriptome reprogramming during PTI and ETI. The data presented clearly shows a major requirement of RNL towards TNL-triggered transcriptome reprogramming and partial requirement in CNL-triggered transcriptome reprogramming. However, RNLs play a limited role in transcriptional reprogramming during basal resistance. This is somewhat surprising given ADR1s and NRG1s role in basal resistance (Fig. 3). 

Text and conclusions are difficult follow with respect to transcriptome data. Most of the data shown in figure is based on number of genes that are differentially regulated in wild type and different mutants. Some details on what kinds of genes are there in these gene sets will help readers to appreciate the data. Supplementary Table S2 and S3 fails to open when downloaded; it says unsupported format.

S2 Figure data is not presented in the text.

---

## [Decision Letter · Decision Letter 2]

27 May 2020

Dear Dr El Kasmi,

Thank you for submitting your revised Research Article entitled "Arabidopsis defense activation by sensor NLRs requires two unequally redundant helper NLR families acting as classical CNLs" for publication in PLOS Biology. I have now obtained advice from original reviewer 1 and the Academic Editor, who assessed your response to the comments from reviewers 2-4. 

Based on the reviews, we will probably accept this manuscript for publication, assuming that you will modify the manuscript to address the remaining points raised by reviewer 1. You will also need to improve the presentation of the error bars in Figure 1B and Figure 2B, which is currently confusing. Please also make sure to address the data and other policy-related requests noted at the end of this email.

We expect to receive your revised manuscript within two weeks. Your revisions should address the specific points made by reviewer 1. In addition to the remaining revisions and before we will be able to formally accept your manuscript and consider it "in press", we also need to ensure that your article conforms to our guidelines. A member of our team will be in touch shortly with a set of requests. As we can't proceed until these requirements are met, your swift response will help prevent delays to publication.

*Copyediting*

*Published Peer Review History*

*Early Version*

*Submitting Your Revision*

Sincerely,

Di Jiang, PhD

PLOS Biology

DATA POLICY:

Regardless of the method selected, please ensure that you provide the individual numerical values that underlie the summary data displayed in the following figure panels as they are essential for readers to assess your analysis and to reproduce it: Figures 1AB, 2BC, 3ABCD, 4AB, 5BD, S1, S2ABFG, S3, S4, S5A-D, S6AB. NOTE: the numerical data provided should include all replicates AND the way in which the plotted mean and errors were derived (it should not present only the mean/average values).

Reviewer remarks:

Reviewer #1 (Johannes Stuttmann, signed review): 

Saile et al., revised manuscript:

In the revised version and also the comments to the reviewers, the authors deal with all critical points in a (more than) satisfactory manner. The manuscript has substantially improved, and provides a very thorough and detailed analysis of RNL-mediated defence responses - which adds important details to our current knowledge.

There are just a few minor points I would like to mention:

lines 70-72: The current wording may imply that all CNLs/TNLs form the resistosome. Indeed, this was shown only for ZAR1, although data suggest it may apply to additional CNLs, and the situation is less clear for TNLs. Consider rephrasing.

Line 486 and following: "In N. benthamiana, NbNRG1 might play a more significant role, because it is required for cell death induction downstream of all tested sensor TNLs or TIR domains (S2 Table, [27])."

Reference [27] is Lapin et al., which is inappropriate here; no Nbnrg1 lines were anylzed in this work. Castel et al NPH and Qi et al PNAS should be cited here.

Line 490 and following: Gantner et al. does not include data on NRG1-dependency of TIR/TNLs (only NbSAG101b), also Qi 2018 PNAS should be cited here.

Line 592 and following: It is odd to see Jane Parker acknowledged for providing this eds1-12 line when it does not originate from her lab.

---

## [Editor Report · Decision Letter 3]

17 Aug 2020

Dear Dr El Kasmi,

On behalf of my colleagues and the Academic Editor, Dr. Xinnian Dong, I am pleased to inform you that we will be delighted to publish your Research Article in PLOS Biology. 

Early Version

PRESS 

Thank you again for submitting your manuscript to PLOS Biology and for your support of Open Access publishing. Please do not hesitate to contact us if we can provide any assistance during the production process.

Kind regards,

Pamela Berkman

Publishing Editor, 

PLOS Biology

on behalf of

Ines Alvarez-Garcia,

Senior Editor

PLOS Biology